# GenPose: Generative Category-level Object Pose Estimation via Diffusion Models

**Jiyao Zhang**[1,2,3 *] **, Mingdong Wu**[1,3 *] **, Hao Dong**[1,3 †]

[1] Center on Frontiers of Computing Studies, School of Computer Science, Peking University
[2] Beijing Academy of Artificial Intelligence
[3] National Key Laboratory for Multimedia Information Processing,
School of Computer Science, Peking University
jiyaozhang@stu.pku.edu.cn, {wmingd, hao.dong}@pku.edu.cn

## Abstract

Object pose estimation plays a vital role in embodied AI and computer vision, enabling intelligent agents to comprehend and interact with their surroundings. Despite the practicality of category-level pose estimation, current approaches encounter challenges with partially observed point clouds, known as the *multi-hypothesis issue.* In this study, we propose a novel solution by reframing category-level object pose estimation as conditional generative modeling, departing from traditional point-to-point regression. Leveraging score-based diffusion models, we estimate object poses by sampling candidates from the diffusion model and aggregating them through a two-step process: filtering out outliers via likelihood estimation and subsequently mean-pooling the remaining candidates. To avoid the costly integration process when estimating the likelihood, we introduce an alternative method that trains an energy-based model from the original score-based model, enabling end-to-end likelihood estimation. Our approach achieves state-of-the-art performance on the REAL275 dataset and demonstrates promising generalizability to novel categories sharing similar symmetric properties without fine-tuning. Furthermore, it can readily adapt to object pose tracking tasks, yielding comparable results to the current state-of-the-art baselines. Our checkpoints and demonstrations can be found at https://sites.google.com/view/genpose.

## 1 Introduction

Object pose estimation is a fundamental problem in the domains of robotics and machine vision, providing a high-level representation of the surrounding world. It plays a crucial role in enabling intelligent agents to understand and interact with their environments, supporting tasks such as robotic manipulation, augmented reality, and virtual reality [1; 2; 3]. In the existing literature, category-level object pose estimation [4; 5; 6] has emerged as a more practical approach compared to instance-level pose estimation since the former eliminates the need for a 3D CAD model for each individual instance. The main challenge of this task is to capture the general properties while accommodating the intra-class variations, *i.e.*, variations among different instances within a category [7; 5; 8].

Figure 1: *Multi-hypothesis issue* in category-level object pose estimation comes from (a) symmetric objects and (b) partial observation.

---

*: equal contribution, †: corresponding author

To address this task, previous studies have explored regression-based approaches. These approaches involve either recovering the object pose by predicting correspondences between the observed point cloud and synthesized canonical coordinates [4; 7], or estimating the pose in an end-to-end manner [9; 6]. The former approaches leverage category prior information, such as the mean point cloud, to handle intra-class variations. As a result, they have demonstrated strong performance across various metrics. On the other hand, the latter approaches do not rely on the coordinate mapping process, which inherently makes them faster and amenable to end-to-end training.

Despite the promising results of previous approaches, a fundamental issue inherent to regression-based training persists. This issue, known as the *multi-hypothesis problem*, arises when dealing with a partially observed point cloud. In such cases:

> *Multiple feasible pose hypotheses can exist, but the network can only be supervised for a single pose due to the regression-based training.*

Figure 1(a) illustrates this problem by showcasing the feasible ground truth poses of symmetric objects (*e.g.*, bowls). Moreover, partially observed point clouds of an object may appear similar from certain views (*e.g.*, a mug with an obstructed handle may look the same from certain views, as shown in Figure 1(b)), further exacerbating the multiple hypothesis issue. Previous studies have proposed ad-hoc solutions to tackle this issue, such as designing special network architectures [10; 11] or augmenting the ground truth poses [4; 5] for symmetric objects. However, these approaches cannot fundamentally resolve this issue due to the lack of generality. Consequently, an ideal object pose estimator should model a pose distribution instead of regressing a single pose when presented with a partially observed point cloud.

To this end, we formulate category-level object pose estimation as conditional generative modeling that inherently models the distribution of multiple pose hypotheses conditioned on a partially observed point cloud. To model the conditional pose distribution, we employ score-based diffusion models, which have demonstrated promising results in various conditional generation tasks [12]. Specifically, we estimate the score functions, *i.e.*, the gradients of the log density, of the conditional pose distribution perturbed by different noise levels. These trained gradient fields can provide guidance to iteratively refine a pose, making it more compatible with the observed point cloud. In the testing phase, we estimate the object pose of a partial point cloud by sampling from the conditional pose distribution using an MCMC process incorporated with the learned gradient fields. As a result, our method can propose multiple pose hypotheses for a partial point cloud, thanks to stochastic sampling.

However, it is essential to note that a sampled pose might be an outlier of the conditional pose distribution, meaning it has a low likelihood, which can severely hurt the performance. To address this issue, we sample a group of pose candidates and then filter out the outliers based on the estimated likelihoods. Unfortunately, estimating the likelihoods from score-based models requires a highly time-consuming integration process [13], rendering this approach impractical. To overcome this challenge, we propose an alternative solution that trains an energy-based model from the original score-based model for likelihood estimation. Following training, the energy network is guaranteed to estimate the log-likelihood of the original data distribution up to a constant. By ranking the candidates according to the energy network's output, we can filter out candidates with low ranks (*e.g.*, the last 40%). Finally, the output pose is aggregated by mean-pooling the remaining pose candidates.

Our experiments demonstrate several superiorities over the existing methods. Firstly, without any ad-hoc network or loss design for symmetry objects, our method achieves state-of-the-art performance and surpasses 50% and 60% on the strict $5°2cm$ and $5°5cm$ metrics, respectively, on the REAL275 dataset. Besides, our method can directly generalize to novel categories without any fine-tuning to some degree due to its prior-free nature. Our methods can be easily adapted to the object pose tracking task with few modifications and achieve comparable performance with the SOTA baselines.

Our contributions are summarized as follows:

- We study a fundamental problem, namely *multi-hypothesis issue*, that has existed in category-level object pose estimation for a long time and introduce a generative approach to address this issue.
- We propose a novel framework that leverages the energy-based diffusion model to aggregate the candidates generated by the score-based diffusion model for object pose estimation.
- Our framework achieves exceptionally state-of-the-art performance on existing benchmarks, especially on symmetric objects. In addition, our framework is capable of object pose tracking and can generalize to objects from novel categories sharing similar symmetric properties.

## 2 Related Works

### 2.1 Category Level Object Pose Estimation

Category-level object pose estimation [4; 14; 5; 15] aims to predict pose of various instances within the same object category without requiring CAD models. To this end, NOCS [4] introduce normalized object coordinate space for each object category and predicts the corresponding object shape in canonical space. Object pose is then calculated by pose fitting using the Umeyama [16] algorithm. CASS [17], DualPoseNet [9], and SSP-Pose [18] end-to-end regress the pose and simultaneously reconstruct the object shape in canonical space to improve the precision of pose regression. SAR-Net [10] and GPV-Pose [11] specifically design symmetric-aware reconstruction to enhance the performance of predicting the translation and size of symmetrical objects while regressing the object pose. However, direct regression of pose or object coordinates in canonical space fail when there is considerable intra-class variation. To overcome this, FS-Net [6] introduces an online box-cage-based 3D deformation augmentation method, and RBP-Pose [19] proposes a nonlinear data augmentation method. On the other hand, SPD [7], SGPA [5], CR-Net [20], and DPDN [8] introduce category-level object priors (*e.g.*, mean shape) and learn how to deform the prior to get the object coordinates in canonical space. These methods have demonstrated strong performance. Overall, the existing category-level object pose estimation methods are regression-based. However, considering symmetric objects (*e.g.*, bowls, bottles, and cans) and partially observed objects (*e.g.*, mugs without visible handles), there are multiple plausible hypotheses for object pose. Therefore, the problem of object pose estimation should be formulated as a generation problem rather than a regression problem.

### 2.2 Score-based Generative Models

In the realm of estimating the gradient of the log-likelihood pertaining to specified data distribution, a pioneering approach known as the score-based generative model [21; 22; 23; 24; 25; 13; 26] was originally introduced by [22]. In an effort to provide a feasible substitute objective for score-matching, the denoising score-matching (DSM) technique[21] has further put forward a viable proposition. To enhance the scalability of the score-based generative model, [23] has introduced a sliced score-matching objective, which involves projecting the scores onto random vectors prior to their comparison. They have also presented annealed training for denoising score matching[24] and have introduced improved training techniques to complement these methods [25]. Additionally, they have extended the discrete levels of annealed score matching to a continuous diffusion process, thereby demonstrating promising results in the domain of image generation [13]. Recent studies have delved further into exploring the design choices of the diffusion process [27], maximum likelihood training [26], and deployment on the Riemann manifold [28]. These advancements have showcased encouraging outcomes when applying score-based generative models to high-dimensional domains, thus fostering their broad utilization across various fields, such as object rearrangement [29], medical imaging [30], point cloud generation [31], scene graph generation [32], point cloud denoising [33], depth completion [34], and human pose estimation [35]. These studies have formulated perception-related problems as either conditional generative modeling or in-painting tasks, thereby harnessing the power of score-based generative models to tackle these challenges. In contrast to these approaches, we incorporate the score-based diffusion model with an additional energy-based diffusion model [36; 37] for filtering out the outliers so as to improve the performance by aggregating the remaining candidates.

## 3 Method

**Task Description:** In this work, we aim to estimate 6D object pose from the partially observed point cloud. The learning agent is given a training set with paired object poses and point clouds $\mathcal{D} = \{(\boldsymbol{p}_i, O_i)\}_{i=1}^n$, where $\boldsymbol{p}_i \in \mathrm{SE}(3)$ and $O_i \in \mathbb{R}^{3 \times N}$ denote a 6D pose and a partially observed 3D point cloud with $N$ points respectively. Given an unseen point cloud $O^*$, the goal is to recover the corresponding ground-truth pose $\boldsymbol{p}^*$.

**Overview:** We formulate the object pose estimation as a conditional generative modeling problem and train a score-based diffusion model $\Phi_\theta$ using the dataset $\mathcal{D}$. Further, an energy-based diffusion model $\Psi_\phi$ is trained from the score-based model $\Phi_\theta$ for likelihood estimation. During test time, given an unseen point cloud $O^*$, we first sample a group of pose candidates $\{\hat{\boldsymbol{p}}_1, \hat{\boldsymbol{p}}_2, ... \hat{\boldsymbol{p}}_K\}$ via the score-based diffusion model $\Phi_\theta$. Then we sort the candidates $\{\hat{\boldsymbol{p}}_i\}_{i=1}^K$ in descending order according

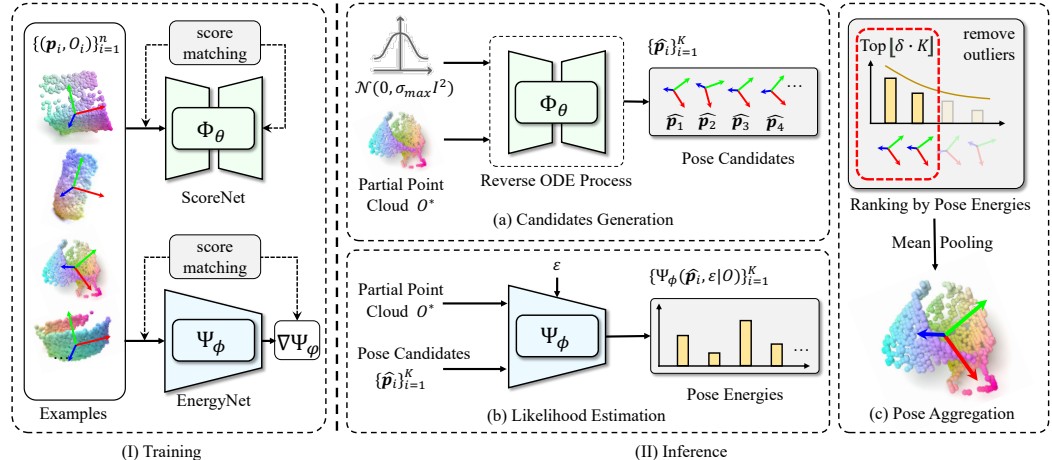

Figure 2: Overview. **(I)** A score-based diffusion model $\mathbf{\Phi}_\theta$ and an energy-based diffusion model $\mathbf{\Psi}_\phi$ is trained via denoising score-matching. **(II)** a) We first generate pose candidates $\{\hat{p}_i\}_{i=1}^K$ from the score-based model and then b) compute the pose energies $\mathbf{\Psi}_\phi(\hat{p}_i, \epsilon|O^*)$ for candidates via the energy-based model. c) Finally, we rank the candidates with the energies and then filter out low-ranking candidates. The remaining candidates are aggregated into the final output by mean-pooling.

to the energy outputs and filter out the last $1 - \delta\%$ candidates (*e.g.*, $\delta = 60\%$). Finally, we aggregate the remaining candidates by mean-pooling to obtain the estimated pose $\hat{p}$.

## 3.1 Sampling Pose Candidates via Score-based Diffusion Model

To address the multi-hypothesis issue, an ideal pose estimator should be capable of being trained on multiple feasible ground truth poses given the same point cloud. Also, the ideal pose estimator should be able to output all possible pose hypotheses of the given point cloud during inference.

To this end, we propose to tackle object pose estimation in a conditional generative modeling paradigm. We assume dataset $\mathcal{D}$ is sampled from an implicit joint distribution $\mathcal{D} = \{(p_i, O_i) \sim p_{\text{data}}(p, O)\}$. We aim to model the conditional pose distribution $p_{\text{data}}(p|O)$ during training, and sample pose hypotheses of an unseen point cloud $O^*$ from $p_{\text{data}}(p|O^*)$ during test-time.

Specifically, we employ a score-based diffusion model [24; 23; 13] to estimate the conditional distribution $p_{\text{data}}(p|O)$. We adopt Variance-Exploding (VE) Stochastic Differential Equation (SDE) proposed by [13] to construct a continuous diffusion process $\{p(t)\}_{t=0}^1$ indexed by a time variable $t \in [0, 1]$ where $p(0) \sim p_{\text{data}}(p|O)$ denotes the ground truth pose of the point cloud $O$. As the $t$ increases from 0 to 1, the time-indexed pose variable $p(t)$ is perturbed by the following SDE:

$$d p = \sqrt{\frac{d[\sigma^2(t)]}{dt}} d\mathbf{w}, \ \sigma(t) = \sigma_{\min}(\frac{\sigma_{\max}}{\sigma_{\min}})^t \tag{1}$$

where $\sigma_{\min} = 0.01$ and $\sigma_{\max} = 50$ are hyper-parameters.

During training, we aim to estimate the *score function* of the perturbed conditional pose distribution $\nabla_p \log p_t(p|O)$ of all $t$, where the $p_t(p|O)$ denotes the marginal distribution of $p(t)$:

$$p_t(p(t)|O) = \int \mathcal{N}(p(t); p(0), \sigma^2(t)\mathbf{I}) \cdot p_0(p(0)|O) \ dp(0) \tag{2}$$

Notably, when $t = 0$, $p_0(p(0)|O) = p_{\text{data}}(p(0)|O)$ is exactly the data distribution.

Thanks to the Denoising Score Matching (DSM) [21], we can obtain a guaranteed estimation of $\nabla_p p_t(p|O)$ by training a score network $\mathbf{\Phi}_\theta : \mathbb{R}^{|\mathcal{P}|} \times \mathbb{R}^1 \times \mathbb{R}^{3 \times N} \to \mathbb{R}^{|\mathcal{P}|}$ via the following objective:

$$\mathcal{L}(\theta) = \mathbb{E}_{t \sim \mathcal{U}(\epsilon, 1)} \left\{ \lambda(t) \mathbb{E}_{\substack{p(0) \sim p_{\text{data}}(p(0)|O), \\ p(t) \sim \mathcal{N}(p(t); p(0), \sigma^2(t)\mathbf{I})}} \left[ \left\| \mathbf{\Phi}_\theta(p(t), t|O) - \frac{p(0) - p(t)}{\sigma(t)^2} \right\|_2^2 \right] \right\} \tag{3}$$

where $\epsilon$ is a hyper-parameter that denotes the minimal noise level. When minimizes the objective in Eq. 3, the optimal score network satisfies $\mathbf{\Phi}_\theta(p, t|O) = \nabla_p \log p_t(p|O)$ according to [21].

After training, we can approximately sample pose candidates $\{\hat{p}_i\}_{i=1}^K$ from $p_{\text{data}}(p|O)$ by sampling from $p_\epsilon(p|O)$, as $\lim_{\epsilon \to 0} p_\epsilon(p|O) = p_{\text{data}}(p|O)$. To sample from $p_\epsilon(p|O)$, we can solve the following *Probability Flow* (PF) ODE [13] where $p(1) \sim \mathcal{N}(\mathbf{0}, \sigma_{\max}^2 \mathbf{I})$, from $t = 1$ to $t = \epsilon$:

$$\frac{dp}{dt} = -\sigma(t)\dot{\sigma}(t)\nabla_p \log p_t(p|O) \tag{4}$$

where the score function $\log p_t(p|O)$ is empirically approximated by the estimated score network $\mathbf{\Phi}_\theta(p, t|O)$ and the ODE trajectory is solved by RK45 ODE solver [38].

In practice, the score network $\mathbf{\Phi}_\theta(p, t|O)$ is implemented without any ad-hoc design for the symmetric objects: The point cloud $O$ is encoded into a global feature by PointNet++ [39], the input pose $p$ is encoded by feed-forward MLPs and the time variable $t$ is encoded by a commonly used projection layer following [13]. The pose $p$ is represented as a 9-D variable $[R|T]$, where $R \in \mathbb{R}^6$ and $T \in \mathbb{R}^3$ denote rotation and translation vectors, respectively. Due to the discontinuity of quaternions and Euler angles in Euclidean space, we employ the continuous 6-D rotation representation $[R_x|R_y]$ following [6; 40]. We defer full details into Appendix A.

### 3.2 Aggregating Pose Candidates via Energy-based Diffusion Model

Though we can sample pose candidates $\{\hat{p}_i\}_{i=1}^K$ from the conditional pose distribution $p_{\text{data}}(p|O)$ via the learned score model $\mathbf{\Phi}_\theta$, we still need to determine a final output estimation $\hat{p} \in \text{SE}(3)$. In other words, the pose candidates need to be aggregated into a single estimation.

An initial approach to aggregating candidates is mean pooling. However, when sampling candidates from $p_{\text{data}}(p|O)$, there is no guarantee that the sampled candidates will have a high likelihood. This means that outlier poses in low-density regions are still likely to be included in the mean-pooled pose, which can negatively impact its performance. Therefore, we propose estimating the data likelihoods $\{p_{\text{data}}(\hat{p}_i|O)\}_{i=1}^K$ to rank the candidates and then filter out low-ranking candidates.

Unfortunately, estimating likelihood via the score model itself requires a time-consuming integration process [13], which is impractical for real-time applications:

$$\log p_{\text{data}}(p|O) \approx \log p_\epsilon(p|O) = \log p_1(p(1)|O) - \frac{1}{2}\int_\epsilon^1 \frac{d[\sigma^2(t)]}{dt}\mathbf{\nabla} \cdot \mathbf{\Phi}_\theta(p(t), t|O)dt \tag{5}$$

where $\{p(t)\}_{t \in [0,1]}$ is the same forward diffusion process as in Eq. 1.

To this end, we propose to train an energy-based model $\mathbf{\Psi}_\phi : \mathbb{R}^{|\mathcal{P}|} \times \mathbb{R}^1 \times \mathbb{R}^{3 \times N} \to \mathbb{R}^1$ that enables an end-to-end surrogate estimation of the data likelihood by supervising the energy-induced gradient $\nabla p\mathbf{\Psi}_\phi(p, t|O)$ with the denoising score-matching objective in Eq. 3:

$$\mathcal{L}(\phi) = \mathbb{E}_{t \sim \mathcal{U}(\epsilon, 1)}\left\{\lambda(t)\mathbb{E}_{\substack{p(0) \sim p_0(p|O), \\ p(t) \sim p_{0t}(p(t)|p(0), O)}}\left[\left\|\nabla_{p(t)}\mathbf{\Psi}_\phi(p(t), t|O) - \frac{p(0) - p(t)}{\sigma(t)^2}\right\|_2^2\right]\right\} \tag{6}$$

In this way, the optimal energy model holds $\nabla_p\mathbf{\Psi}_\phi^*(p, t|O) = \nabla_p \log p_t(p|O)$, or equivalently, $\mathbf{\Psi}_\phi^*(p, t|O) = \log p_t(p|O) + C$ where $C$ is a constant. Although the optimal energy model and the ground truth likelihood differ by a constant $C$, this does not prevent it from functioning as a good surrogate likelihood estimator for ranking the candidates:

$$\mathbf{\Psi}_\phi^*(p_i, \epsilon|O) > \mathbf{\Psi}_\phi^*(p_j, \epsilon|O) \iff \log p_\epsilon(p_i|O) > \log p_\epsilon(p_j|O) \tag{7}$$

Nevertheless, training an energy-based diffusion model from Eq. 6 is known to be difficult and time-inefficient due to the need to calculate the second-order derivations in the objective function. Following [36], we parameterize the energy-based model as $\mathbf{\Psi}_\phi(p, t|O) = \langle p, \mathbf{\Phi}_\phi(p, t|O) \rangle$ to alleviate the training burden. We defer the full training details to Appendix A.

With the trained energy model, we sort the candidates into a sequence $\hat{p}_{\tau_1} \succ \hat{p}_{\tau_2} ... \succ \hat{p}_{\tau_K}$ where:

$$\hat{p}_{\tau_i} \succ \hat{p}_{\tau_j} \iff \mathbf{\Psi}_\phi(\hat{p}_{\tau_i}, \epsilon|O) > \mathbf{\Psi}_\phi(\hat{p}_{\tau_j}, \epsilon|O) \tag{8}$$

Subsequently, we filter out the last $1 - \delta\%$ candidates and obtain $\hat{p}_{\tau_1} \succ \hat{p}_{\tau_2} ... \succ \hat{p}_{\tau_M}$ where $\delta \in (0, 1)$ is a hyper parameter and $M = \lfloor \delta \cdot K \rfloor$.

Finally, we aggregate the remaining candidates $\{\hat{\boldsymbol{p}}_{\tau_i} = (\hat{T}_{\tau_i}, \hat{R}_{\tau_i})\}_{i=1}^M$ by averaging the rotations $\{\hat{R}_{\tau_i}\}_{i=1}^M$ and the translations $\{\hat{T}_{\tau_i}\}_{i=1}^M$ respectively, to obtain the output pose $\hat{\boldsymbol{p}} = (\hat{T}, \hat{R})$. In specific, the translations are pooled by vanilla averaging $\hat{T} = \frac{\sum_{i=1}^M \hat{T}_{\tau_i}}{M}$. To obtain the averaged rotation, we initially translate the rotations into quaternions $\{\hat{\boldsymbol{q}}_{\tau_i}\}_{i=1}^M$. Following [41], the average quaternion can then be found by the following maximization procedure:

$$\hat{\boldsymbol{q}} = \arg\max_{\boldsymbol{q} \in SO(3)} \boldsymbol{q}^T \left( \frac{\sum_{i=1}^M A(\hat{\boldsymbol{q}}_{\tau_i})}{M} \right) \boldsymbol{q}, \ A(\hat{\boldsymbol{q}}_{\tau_i}) = \hat{\boldsymbol{q}}_{\tau_i} \hat{\boldsymbol{q}}_{\tau_i}^T \tag{9}$$

By definition, the solution of Eq. 9 is the eigenvector of the $4 \times 4$ matrix $\frac{\sum_{i=1}^M A(\boldsymbol{q}_{\tau_i})}{M}$ corresponding to the maximum eigenvalue, which can be efficiently solved by QUEST algorithm [42].

### 3.3 Discussion

Despite addressing the multi-hypothesis issue, our method offers several additional advantages:

**No Ad-hoc Design:** Unlike previous approaches, we do not incorporate any ad-hoc designs or tricks into the network architecture for symmetric objects. Both the score and energy models are implemented using commonly used feature extractors (*e.g.*, PointNet++[39]) and feed-forward MLPs. Surprisingly, our method performs exceptionally well in handling symmetric objects (refer to Sec 4.4) and achieves state-of-the-art (SOTA) performance on existing benchmarks (refer to Sec 4.2).

**Prior-Free:** Our method eliminates the requirement of category-level canonical prior, freeing us from designing a shape-deformation module to incorporate the canonical prior. This also provides the potential to generalize our method to objects from unseen categories sharing similar symmetric properties. In Sec 4.4, we present experiments to demonstrate this potential.

**Capable of Pose Tracking:** Thanks to the closed-loop generation process of the diffusion models, we can adapt the single-frame pose estimation framework to pose tracking tasks by warm-starting from the previous predictions. The adapted object pose tracking framework differs from the single-frame estimation framework only in the candidates' sampling. For each frame that receives the point cloud observation $O_{\text{cur}}$, we initialize the PF-ODE in Eq.4 with $\boldsymbol{p}(0.1) \sim \mathcal{N}(\hat{\boldsymbol{p}}_{\text{prev}}, \sigma^2(0.1)\mathbf{I})$, where $\hat{\boldsymbol{p}}_{\text{prev}} \in \text{SE}(3)$ denotes the estimated pose of the previous frame. Subsequently, we solve this modified PF-ODE from $t = 0.1$ to $t = \epsilon$ using the gradient fields $\Phi_\theta(\boldsymbol{p}, t|O_{\text{cur}})$ to sample candidates for pose tracking. By following the same aggregation procedure, we can obtain the pose estimation $\hat{\boldsymbol{p}}_{\text{cur}}$ for the current frame. We summarise the tracking framework in Appendix D. In Sec.4.5, our adapted pose-tracking method outperforms the state-of-the-art object pose-tracking baseline in most metrics.

## 4 Experiments

### 4.1 Experimental Setup

**Datasets.** Our method is trained and evaluated on two common category-level object pose estimation datasets, namely CAMERA and REAL275 [4], following the NOCS [4] convention for splitting the data into training and testing sets. These datasets include 6 daily objects: bottle, bowl, camera, can, laptop, and mug. CAMERA dataset is a synthetic dataset generated using mixed-reality methods. It comprises real background images with synthetic rendered foreground objects and consists of 275K training images and 25K test images. REAL275 dataset is a real-world dataset that employs the same 6 object categories as the CAMERA dataset. It includes 3 unique instances per category in both the training and test sets and consists of 7 scenes for training with 4.3K images and 6 scenes for testing with 2.75K images. Each scene in the REAL275 dataset contains more than 5 objects.

**Metrics.** Following NOCS [4], we report the mean Average Precision (mAP) in $n°$ and $m$ cm to evaluate the accuracy of object pose estimation. Here, $n$ and $m$ denote the prediction error of rotation and translation, respectively, where the predicted rotation error is less than $n°$ and the predicted translation error is less than $m$ cm. Specifically, we use $5°2cm$, $5°5cm$, $10°2cm$, and $10°5cm$ as our evaluation metrics. Similar to NOCS, for symmetric objects (bottles, bowls, and cans), we ignore the

Table 1: **Quantitative comparison of category-level object pose estimation on REAL275 dataset.** We summarize the results reported in the original paper for the baseline method. ↑ represents a higher value indicating better performance, while ↓ represents a lower value indicating better performance. **Data** refers to the format of the input data used by the method, and **Prior** indicates whether the method requires category prior information. '-' indicates that the metrics are not reported in the original paper. **K** represents the number of hypotheses.

| | Method | Data | Prior | $5°2cm$↑ | $5°5cm$↑ | $10°2cm$↑ | $10°5cm$↑ | Parameters(M)↓ |
|---|---|---|---|---|---|---|---|---|
| | NOCS [4] | RGB-D | × | - | 9.5 | 13.8 | 26.7 | - |
| | CASS [17] | RGB-D | × | 19.5 | 23.5 | 50.8 | 58.0 | 47.2 |
| | DualPoseNet [9] | RGB-D | × | 29.3 | 35.9 | 50.0 | 66.8 | 67.9 |
| | SPD [7] | RGB-D | ✓ | 19.3 | 21.4 | 43.2 | 54.1 | 18.3 |
| | CR-Net [20] | RGB-D | ✓ | 27.8 | 34.3 | 47.2 | 60.8 | - |
| | SGPA [5] | RGB-D | ✓ | 35.9 | 39.6 | 61.3 | 70.7 | - |
| Deterministic | DPDN [8] | RGB-D | ✓ | 46.0 | 50.7 | 70.4 | 78.4 | - |
| | FS-Net [6] | D | × | 19.9 | 33.9 | - | 69.1 | 41.2 |
| | GPV-Pose [11] | D | × | 32.0 | 42.9 | 55.0 | 73.3 | - |
| | SAR-Net [10] | D | ✓ | 31.6 | 42.3 | 50.4 | 68.3 | 6.3 |
| | SSP-Pose [18] | D | ✓ | 34.7 | 44.6 | - | 77.8 | - |
| | RBP-Pose [19] | D | ✓ | 38.2 | 48.1 | 63.1 | 79.2 | - |
| | Ours | D | × | **52.1** | **60.9** | **72.4** | **84.0** | **4.4** |
| Probabilistic | Ours($K$=10) | D | × | 71.5 | 75.9 | 85.2 | 90.8 | 2.2 |
| | Ours($K$=50) | D | × | 82.0 | 84.5 | 92.8 | 95.9 | 2.2 |

rotation error around the symmetry axis. For the "mug" category, we consider it a symmetric object when the handle is not visible and a non-symmetric object when the handle is visible.

**Implementation Details.** For a fair comparison, we employed the instance mask generated by MaskRCNN [43] during the inference phase, which was identical to the one used in previous work. Both the Energy-based diffusion model for pose ranking and the score-based diffusion model for pose generation shared the same network structure. The input pointcloud consisted of 1024 points. During the training phase, we used the same data augmentation techniques as FS-Net [6], which are widely adopted in category-level object pose estimation tasks. All experiments were conducted on a single RTX3090 with a batch size of 192. All the experiments are implemented using PyTorch [44].

## 4.2 Comparison with State-of-the-Art Methods

Table 1 provides a comprehensive comparison between our method and the state-of-the-art approaches on the REAL275 [4] dataset, highlighting a significant advancement in performance. In Table 1, "Ours" represents the aggregated pose using $K$ as 50 and $\delta$ as 60%. "Ours($K$=10)" and "Ours($K$=50)" indicate that we set $K$ as 10 and 50, respectively, and select a pose from candidates with the minimum distance to ground truth pose for evaluation. We deal with all categories by a single model.

As shown in Table 1, our approach demonstrates a remarkable improvement over the current SOTA method, GPV-Pose[11], exceeding it by more than 20% in both the rigorous $5°2cm$ and $5°5cm$ evaluation metrics. Significantly, for the first time, we achieve results on the REAL275 dataset, surpassing the impressive thresholds of 50% and 60% in the $5°2cm$ and $5°5cm$ metrics, respectively. These exceptional outcomes further support the efficacy of our approach.

Even when compared to methods that take RGB-D data and category prior as input, our method maintains a notable advantage. Without relying on any additional information, our method achieves a performance boost of over 10% in the metric of $5°5cm$, surpassing the SOTA method, DPDN [8].

In addition, the results that evaluate the pose nearest to the ground truth indicate that condition generative models exhibit tremendous potential for category-level object pose estimation. Another crucial metric is the number of learnable parameters of the network, which significantly impacts the feasibility of model deployment. Table 1 reveals that our method achieves a remarkable accuracy for predicting poses with the fewest network parameters, further improving its deployability.

### 4.3 Ablation Studies

**Number of Pose Candidates and Proportion of Selected Poses.** Using the $10°2cm$ metric, Table 2 demonstrates the impact of two factors on performance: the number of generated pose candidates $M$, and the proportion $\delta$ of selected poses out of the total pose candidates.

The results show a notable improvement when $M$ increases from 10 to 50. This enhancement can be attributed to the increasing number of sampling times, resulting in a pose candidate set that aligns more closely with the predicted distribution. However, the improvement becomes marginal when $M$ is further increased from 50 to 100. This can be attributed to the predictions that approach the upper limit of the aggregation approach.

Table 2: Ablation on number of pose candidates $K$ and proportion of selected poses $\delta$.

| $K$ \ $\delta$ | 20% | 40% | 60% | 80% | 100% |
|---|---|---|---|---|---|
| 10 | 65.3 | 67.7 | 67.9 | 67.7 | 65.0 |
| 50 | 70.2 | 71.8 | 72.4 | 71.9 | 69.7 |
| 100 | **70.8** | **72.2** | **72.6** | **72.5** | **70.2** |

After considering the trade-off between performance and overhead, we ultimately decided to adopt $K = 50$. In view of the inherent challenge in training an energy model that precisely aligns with the actual distribution, it becomes apparent that employing a small delta value, as demonstrated in Table 2, does not yield an optimal outcome. Consequently, we regard the energy model primarily as a detector for outliers. The most successful results of this paper were achieved using a delta value of 60%, which effectively mitigated the presence of relatively insignificant noise originating from the sampling process.

**Effectiveness of the Energy-based Likelihood Estimator $\Psi_\phi$.** In this part, we report the results of three distinct ranking methodologies, namely "Random", "Energy", and "GT", employed to sort pose candidates $\{\hat{\boldsymbol{p}}_i\}_{i=1}^K$. "Random" applies a stochastic shuffling to $\{\hat{\boldsymbol{p}}_i\}_{i=1}^K$, providing a baseline approach. "Energy" uses our trained energy model to rank $\{\hat{\boldsymbol{p}}_i\}_{i=1}^K$. "GT" ranks $\{\hat{\boldsymbol{p}}_i\}_{i=1}^K$ in ascending order of the distances to the ground truth pose, serving as an upper limit for all ranking methods.

The absence of mean pooling in Table 3 implies that one pose candidate is chosen randomly for evaluation. We observe that utilizing an energy-based likelihood estimator as a sampler enhances the sort-

Table 3: Effectiveness of the energy-based likelihood estimator $\Psi_\phi$.

| Ranking | Mean pooling | $5°2cm$ | $5°5cm$ | $10°2cm$ | $10°5cm$ |
|---|---|---|---|---|---|
| Random | × | 13.5 | 49.1 | 21.5 | 76.3 |
| Random | ✓ | 49.4 | 58.6 | 68.5 | 80.7 |
| Energy | ✓ | **52.1** | **60.9** | **72.4** | **84.0** |
| GT(upper bound) | ✓ | 62.1 | 66.8 | 80.9 | 86.9 |

ing of pose candidates and effectively eliminates outliers. This improvement is significant compared to random sampling. Although doesn't reach the upper bound, we offer a novel and practical approach for ranking and aggregating poses sampled from estimated distribution and demonstrate its potential.

To better understand the energy model's impact, we further investigated the correlation between pose error and energy output. As depicted in Figure 3, there is a general negative correlation between the output and the error of the energy model. However, the energy model excels at distinguishing poses with significant error differences but performs poorly when distinguishing poses with low errors. For instance, the energy model assigns similar values (*e.g.*, translation, $<2cm$) or even incorrect values (*e.g.*, rotation, $<10°$) for poses with low errors. Additionally, the energies associated with low-error poses (*e.g.*, $<10°$, $<2cm$) are higher than those of high-error poses (*e.g.*, $>20°$, $>4cm$).

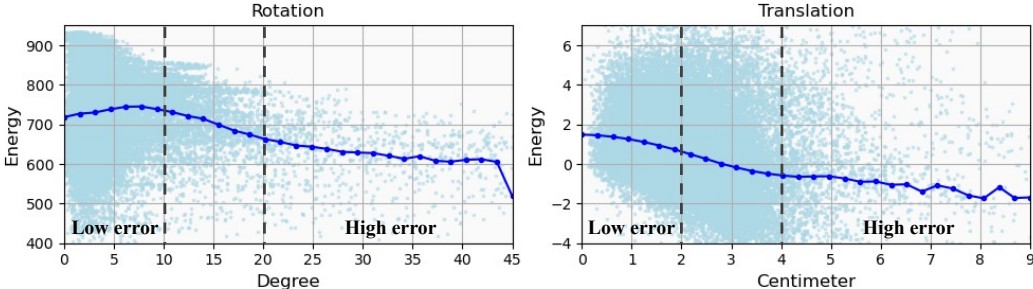

Figure 3: **Energy-Error Correlation Analysis.** Points for raw scatters plot, curves for mean error.

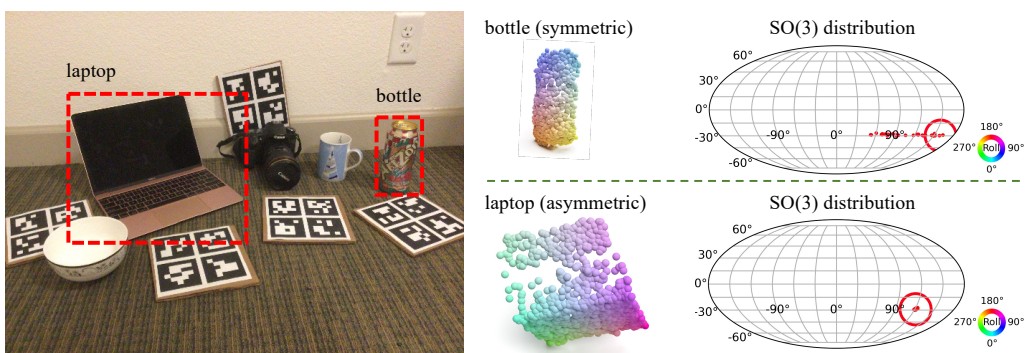

Figure 4: **The predicted conditional rotation distribution of symmetric and asymmetric objects.** The left figure is an example from the REAL275 dataset, while the right figure shows the distribution of the generated rotations for the symmetrical bottle and the asymmetric laptop. The rotation distribution is visualized by plotting yaw as latitude, pitch as longitude, and roll as the color, following the approach inspired by [45; 46]. The ground truth is represented by the center of the open circle, and the dot shows the result of generating rotations 50 times.

## 4.4 Analysis on Symmetric Objects

**Performance.** Our approach outperforms the state-of-the-art method RBP-Pose [19], especially for symmetrical objects, as shown in Table 4. Importantly, our method does not rely on specific network structures or loss designs for these categories. Notably, for 'mug' category, where asymmetry is present when the handle is visible and symmetry when it is not, our method effectively addresses this challenge and achieves significant performance improvements compared to previous methods. To visually depict the disparity between symmetric and asymmetric objects, Figure 4 visually demonstrates the accuracy of our method in predicting rotation distributions for both symmetric and asymmetric objects.

Table 4: Per-category results of our method and RBP-Pose.

| Category | Symmetric | 5°2cm | | 5°5cm | |
|---|---|---|---|---|---|
| | | RBP-Pose | Ours | RBP-Pose | Ours |
| camera | ✕ | 1.3 | **2.9** | 1.5 | **3.2** |
| laptop | | 41.3 | **63.4** | 75.2 | **91.4** |
| average | | 21.3 | **32.2**(11.9↑) | 38.4 | **47.3**(8.9↑) |
| bottle | ✓ | 38.7 | **52.6** | 43.5 | **60.9** |
| bowl | | 75.4 | **85.4** | 81.7 | **92.6** |
| can | | 53.5 | **72.5** | 67.1 | **80.4** |
| average | | 55.9 | **70.2**(14.3↑) | 64.1 | **78.0**(13.9↑) |
| mug | - | 18.9 | **35.7**(16.8↑) | 19.4 | **36.4**(17.0↑) |

**Toward Cross-category Object Pose Estimation.** Our method demonstrates a surprisingly cross-category generalizability, as it directly predicts the distribution of object poses, eliminating the reliance on object category prior. Table 5 verifies the remarkable ability of our approach to generalize across categories sharing similar symmetric properties.

Our approach demonstrates consistent and robust performance when tested on unseen categories with similar symmetrical attributes, whereas the baseline performance exhibits a significant decline. We hypothesize that the specific out-of-distribution (OOD) generalization ability of our method arises from the learned feature space of the point cloud. For example, although the bowl

Table 5: **Cross category evaluation on REAL275.** The left and right sides of '/' respectively indicate the performance when the testing category is seen and unseen in training data.

| Category | Method | 5°2cm | 5°5cm | 10°2cm | 10°5cm |
|---|---|---|---|---|---|
| bowl | SAR-Net[10] | 58.1/36.4 | 66.0/47.3 | 83.7/59.4 | 93.6/81.5 |
| | RBP-Pose[19] | 75.4/0.0 | 81.7/6.9 | 92.1/0.1 | 100.0/30.7 |
| | Ours | **85.4/64.5** | **92.6/72.5** | **93.1/87.2** | **100.0/98.6** |
| bottle | SAR-Net[10] | 43.5/11.7 | 54.0/23.0 | 61.3/33.6 | 79.8/68.0 |
| | RBP-Pose[19] | 38.7/4.3 | 43.5/5.8 | 76.4/24.7 | 89.8/29.7 |
| | Ours | **52.6/39.0** | **60.9/53.2** | **81.4/73.6** | **92.7/94.6** |
| can | SAR-Net[10] | 32.2/7.3 | 62.2/52.3 | 52.5/12.1 | 92.9/87.9 |
| | RBP-Pose[19] | 53.5/0.8 | 67.1/21.0 | 78.8/2.6 | 96.3/61.7 |
| | Ours | **72.5/62.5** | **80.4/74.0** | **88.8/81.6** | **99.8/99.7** |

category is considered OOD, the extracted features from point clouds in the bowl category may exhibit similarities or closeness to seen categories, to some extent. In other words, a bowl may share visual characteristics with items such as cans, bottles, or mugs, as identified by the PointNet++ of the

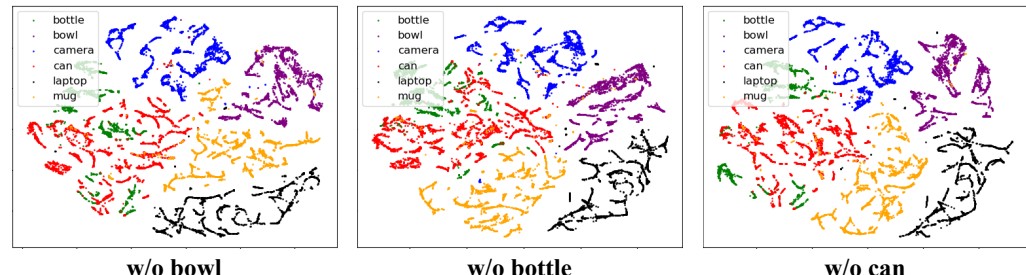

**Figure 5: Visualization of the features.** 'w/o' denotes the unseen category during training process.

ScoreNet. To validate this hypothesis, we conducted a t-SNE [47] analysis on the point cloud feature space of the ScoreNet. Specifically, we extracted features from the point clouds of objects in the seen and unseen test set and visualized the t-SNE results. As shown in Figure 5, the results demonstrate that features from cans and bottles tend to intermingle, aligning with the accurate observation that both cans and bottles exhibit symmetrical cylindrical shapes. Meanwhile, features from the bowl category show proximity to features from mugs.

### 4.5 Category-level Object Pose Tracking

Table 6 shows the performance of category-level object pose tracking of our tracking method and baselines on REAL275 datasets. For the first frame, we utilized a perturbed ground truth pose as the initial object pose, following CAPTRA [48] and CATRE [49]. Despite being directly transferred from a single-image prediction method, our approach has achieved performance comparable to the state-of-the-art method, CATRE. Remarkably, we significantly outperform CATRE in the metrics of $5°5cm$ and mean rotation error. This highlights the expansibility of our pose estimation method for more nuanced downstream tasks. Despite our lower FPS compared to CATRE, our method still adequately fulfills usage requirements, particularly for robot operations.

Table 6: **Results of category-level object pose tracking on REAL275.** The results are averaged over all 6 categories. The best performance is in **bold** and the second best is underscored.

| Method | Oracle ICP[48] | 6-PACK[50] | iCaps[51] | CAPTRA[48] | CATRE[49] | Ours |
|---|---|---|---|---|---|---|
| Input | RGBD | RGBD | RGBD | D | D | D |
| Init. | GT. | GT. Pert. | Det. and Seg. | GT. Pert. | GT. Pert. | GT. Pert. |
| Speed(FPS)↑ | - | 10 | 1.84 | 12.66 | **89.21** | 17.18 |
| 5°5cm↑ | 0.7 | 33.3 | 31.6 | 62.2 | 57.2 | **71.5** |
| $R_{err}$(°)↓ | 40.3 | 16.0 | 9.5 | 5.9 | 6.8 | **4.2** |
| $t_{err}$(cm)↓ | 7.7 | 3.5 | 2.3 | 7.9 | **1.2** | 1.5 |

## 5 Conclusion and Discussion

In this work, we study a fundamental problem, namely the *multi-hypothesis issue*, that has existed in category-level object pose estimation for a long time. To overcome this issue, we propose a novel object pose estimation framework that initially generates the pose candidates via a score-based diffusion model. To aggregate the candidates, we leverage an energy-based diffusion model to filter out the outliers, which avoids the costly integration process of the score-based model. Our framework achieves exceptionally state-of-the-art performance on existing benchmarks, especially on symmetric objects. In addition, our framework is capable of object pose tracking and can generalize to objects from novel categories sharing similar symmetric properties.

**Limitations and Future Works:** Although our framework achieves real-time object pose tracking at approximately 17 FPS, the efficiency of single-frame object pose estimation is still limited by the costly sampling process of the score-based model. In the future, we may leverage the recent advances in accelerating the sampling process of the diffusion models to speed up the inference [52; 53]. In addition to passively filtering out the outliers, we may incorporate reinforcement learning to train an agent that actively increases the likelihood of the estimated pose [54].

**Boarder Impact:** This work opens the door to leveraging energy-based diffusion models to improve the generation results, which might facilitate more research in border communities.

## Acknowledgments and Disclosure of Funding

This work is supported by the National Natural Science Foundation of China - General Program (Project ID: 62376006), National Youth Talent Support Program (Project ID: 8200800081), and Beijing Municipal Science & Technology Commission (Project ID: Z221100003422004).

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

# A  Implementation Details

**Architecture of the Score Network**    The detailed architecture of the score network $\Phi_\theta$ is illustrated in Figure 6. We utilize PointNet++ [39] to extract the global geometry feature $\mathcal{F}_O$ of the partially observed point cloud $O^*$. And the sampled pose $p$ and timestep $t$ features are embedded as $\mathcal{F}_p$ and $\mathcal{F}_t$, respectively, using a Multi-Layer Perceptron (MLP). Then $\mathcal{F}_O$, $\mathcal{F}_p$ and $\mathcal{F}_t$ are concatenated to obtain the global feature $\mathcal{F}$, and three parallel branches are employed to predict the scores of $R_x$, $R_y$, and $T$ individually, where $[R_x|R_y] \in \mathbb{R}^6$ and $T \in \mathbb{R}^3$ denote rotation and translation vectors, respectively. And $[R_x|R_y]$ is a continuous rotation representation proposed by [40] to address the discontinuity of quaternions and Euler angles in Euclidean space. As introduced in [40], the mapping from SO(3) to the 6D representation of rotation is:

$$g_{GS}\left([\mathbf{a_1} \quad \mathbf{a_2} \quad \mathbf{a_3}]\right) = [\mathbf{a_1} \quad \mathbf{a_2}] \tag{10}$$

The mapping form the 6D representation to SO(3) is:

$$f_{GS}\left([\mathbf{a_1} \quad \mathbf{a_2}]\right) = [\mathbf{b_1} \quad \mathbf{b_2} \quad \mathbf{b_3}] \tag{11}$$

$$b_i = \begin{bmatrix} \begin{cases} N(\mathbf{a_1}) & \text{if } i = 1 \\ N(\mathbf{a_2} - (\mathbf{b_1} \cdot \mathbf{a_2})\mathbf{b_1}) & \text{if } i = 2 \\ \mathbf{b_1} \times \mathbf{b_2} & \text{if } i = 3 \end{cases} \end{bmatrix} \tag{12}$$

Here $N(\cdot)$ denotes a normalization function.

**Architecture of the Energy Network**    The energy network $\Psi_\phi$ shares exactly the same architecture with the score network $\Phi_\theta$. The inputs are first fed into $\Phi_\phi$ to obtain a score-shaped vector $\Phi_\phi(p, t|O) \in \mathbb{R}^{|\mathcal{P}|}$. Then, the output energy is calculated by the dot product between the input pose and the score-shaped vector $\Psi_\phi(p, t|O) = \langle p, \Phi_\phi(p, t|O) \rangle \in \mathbb{R}^1$.

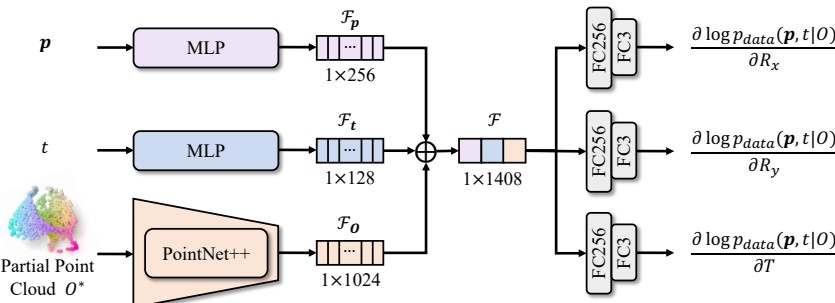

Figure 6: **Architecture of the score network $\Phi_\theta$.** $p$ denotes sampled 6D object poses. $O^*$ denotes partially observed 3D point cloud condition. $t$ denotes timestep. $\oplus$ denotes the concatenation operator.

# B  Qualitative Comparison on REAL275

Figure 2 illustrates the qualitative comparison results between our method and RBP-Pose [19] on the REAL275 dataset. The images are accompanied by red boxes highlighting objects that exhibit noticeable differences in the predicted results. Additionally, the bottom-right corner of each image provides an enlarged view of the highlighted object, showing the ground truth pose as well as the poses estimated by RBP-Pose and our approach. Our method demonstrates a significant performance improvement compared to RBP-Pose, particularly in the case of objects such as mugs. Notably, in the fourth row of the figure, it can be observed that our method achieves highly accurate poses even when only a small portion of the mug handle is visible. This success can be attributed to the fact that, during the training process, a unique pose exists when the mug handle is visible. However, when the mug handle becomes occluded, a multi-hypothesis problem arises, which our generative formulation effectively handles.

# C  More Results and Analysis

## C.1  Per-category Results

Figure 8 demonstrates a quantitative comparison between our method and the state-of-the-art depth-based approach, RBP-Pose [19], for various object categories at different thresholds. The results

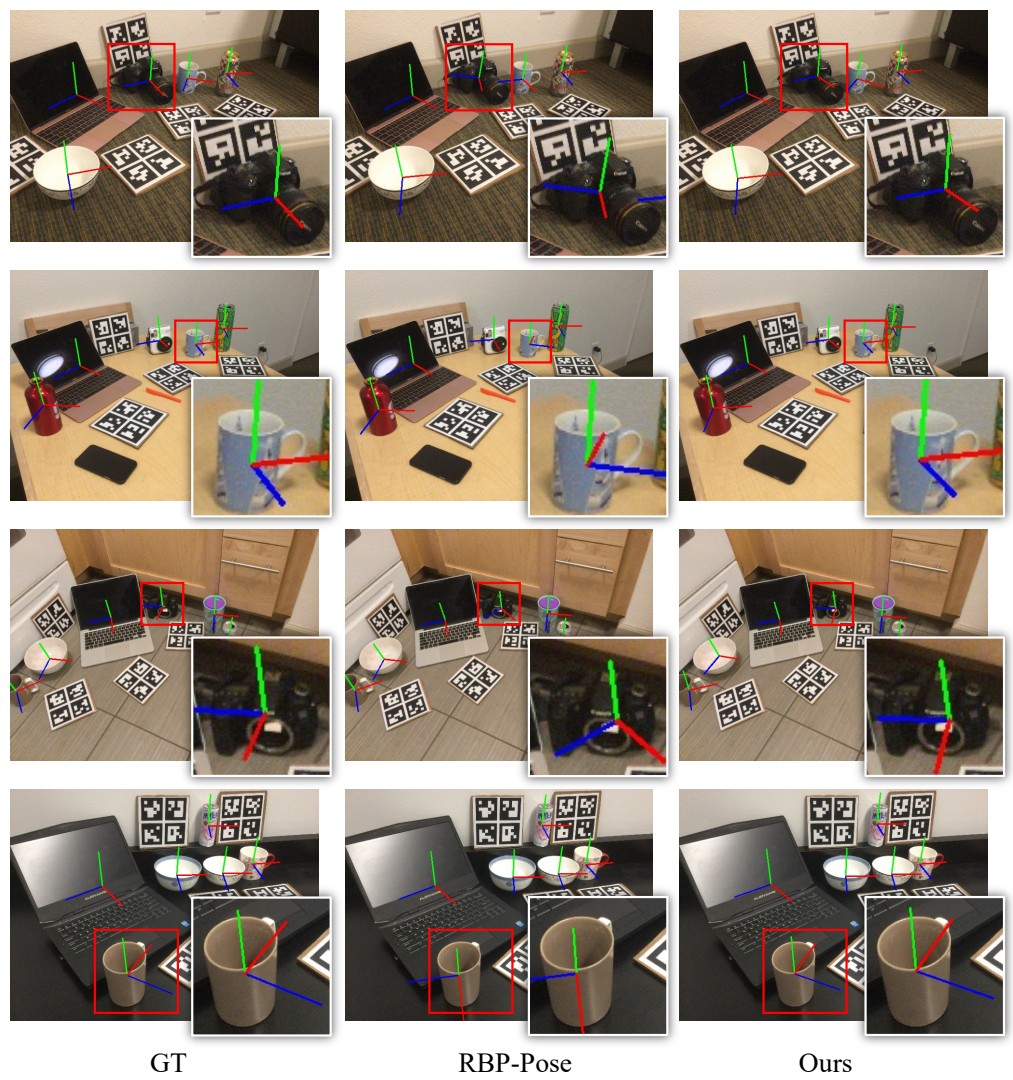

| GT | RBP-Pose | Ours |

Figure 7: **Qualitative comparison with RBP-Pose [19] on REAL275.** The left column represents the ground truth pose, the middle column represents the results of RBP-Pose, the right column represents the results of our approach.

clearly indicate that our method outperforms RBP-Pose in all metrics, despite the fact that we do not incorporate augmentation specifically designed for symmetric objects during the training phase, unlike RBP-Pose. Our approach exhibits significant improvements, particularly in regions with stringent threshold requirements. This emphasizes the superior performance of our generative category-level object 6D pose estimation approach in effectively addressing the multi-hypothesis challenges posed by symmetric objects and partial observations, thereby enabling its successful application in robot manipulation tasks demanding precise object pose prediction.(*e.g*., pouring liquids.)

## C.2    Results on CAMERA

Table 7 illustrates a quantitative comparison between our method and the baselines on the CAM-ERA [4] dataset. The results clearly demonstrate the remarkable performance enhancement achieved by our method. When compared to approaches that rely solely on depth data as network input, as well as those that utilize RGB-D and shape priors as network input, our method consistently outperforms them, surpassing the current state-of-the-art performance. Notably, our method exhibits a particularly pronounced advantage when stricter accuracy requirements are imposed, such as the $5°2cm$ metric. In this case, our method outperforms the current SOTA method, RBP-Pose, by an impressive margin of 6.4%. This significant improvement highlights the efficacy of our approach.

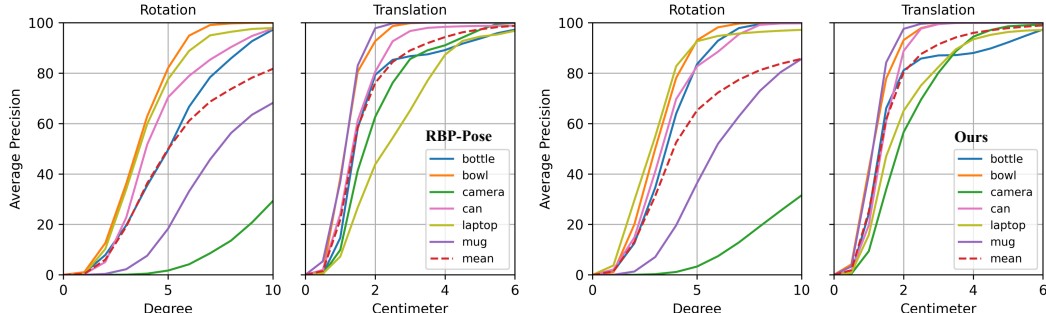

Figure 8: **Per-category quantitative comparison with RBP-Pose [19] on REAL275.** The left represents the results of RBP-Pose, while the right represents the results of our approach.

Table 7: **Quantitative comparison of category-level object pose estimation on CAMERA [4] dataset.** We summarize the results reported in the original paper for the baseline method. ↑ represents a higher value indicating better performance, while ↓ represents a lower value indicating better performance. **Data** refers to the format of the input data used by the method, and **Prior** indicates whether the method requires category prior information. '-' indicates that the metrics are not reported in the original paper. **K** represents the number of hypotheses."

| | Method | Data | Prior | 5°2cm↑ | 5°5cm↑ | 10°2cm↑ | 10°5cm↑ | Parameters(M)↓ |
|---|---|---|---|---|---|---|---|---|
| | NOCS [4] | RGB-D | × | 32.3 | 40.9 | 48.2 | 64.6 | - |
| | DualPoseNet [9] | RGB-D | × | 64.7 | 70.7 | 77.2 | 84.7 | 67.9 |
| | SPD [7] | RGB-D | ✓ | 54.3 | 59.0 | 73.3 | 81.5 | 18.3 |
| | CR-Net [20] | RGB-D | ✓ | 72.0 | 76.4 | 81.0 | 87.7 | - |
| | SGPA [5] | RGB-D | ✓ | 70.7 | 74.5 | 82.7 | 88.4 | - |
| Deterministic | GPV-Pose [11] | D | × | 72.1 | 79.1 | - | 89.0 | - |
| | SAR-Net [10] | D | ✓ | 66.7 | 70.9 | 75.3 | 80.3 | 6.3 |
| | SSP-Pose [18] | D | ✓ | 64.7 | 75.5 | - | 87.4 | - |
| | RBP-Pose [19] | D | ✓ | 73.5 | 79.6 | 82.1 | 89.5 | - |
| Probabilistic | Ours | D | × | **79.9** | **84.4** | **84.6** | **89.6** | **4.4** |
| | Ours($K$=10) | D | × | 90.8 | 93.0 | 93.4 | 95.7 | 2.2 |
| | Ours($K$=50) | D | × | 95.5 | 96.4 | 97.2 | 98.2 | 2.2 |

## C.3 Real World Experiments

We have also successfully integrated our approach with robot manipulation capabilities, as demonstrated through various experiments conducted with the UFACTORY xArm6 equipped with RealSense D435. **The demonstrations can be found in the supplementary video or on the project website.** As shown in Figure 9, we illustrate the following three tasks:

**Pouring Task.** This task involves transferring the contents (*e.g.*, water) from one container to another. The demonstration highlights the potential of combining our approach with heuristic strategies, enabling functional robot operations.

**Stacking Task.** In this task, we focused on piling up objects of the same category, like organizing scattered bowls on a tabletop. This demonstrates the precision of the estimated object pose, as accurate knowledge of object poses is crucial for completing this task.

**Handover Task.** This task involved either receiving objects from human hands to perform tasks or passing objects to person. The demonstration exemplified one form of human-robot interaction empowered by our method.

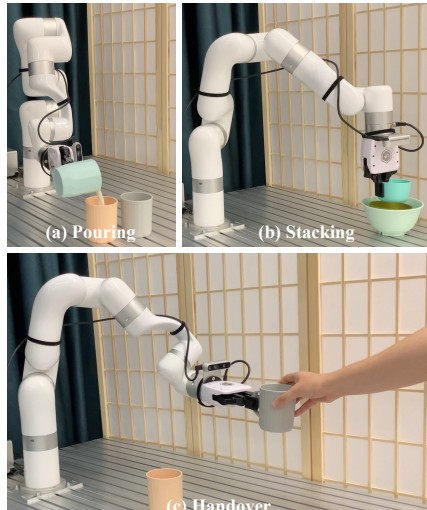

Figure 9: **Pose estimation for robot manipulation tasks.** We demonstrate three types of tasks.

# D  Details of Object Pose Tracking

Our pose estimation framework can be adapted to pose tracking with minor modifications. With the learned score network $\mathbf{\Phi}_\theta$, the tracking algorithm is summarised as follows:

---

**Algorithm 1** Our Object Pose Tracking Framework

---

1: **Initialisation:** Score network $\mathbf{\Phi}_\theta$, energy network $\mathbf{\Psi}_\phi$, initial point cloud $O_0$, the initial ground truth pose reference $\boldsymbol{p}_0$ and the tracking horizon $T$.
2: **for** $t = 1$ **to** $T$ **do**
3:     Receive current observation $O_t$
4:     $\boldsymbol{z}_1, \boldsymbol{z}_2, ... \boldsymbol{z}_K \sim \mathcal{N}(\boldsymbol{p}_{t-1}, 0.1^2 I)$        ▷ Sample initial poses around the previous prediction.
5:     Sample candidates $\{\hat{\boldsymbol{p}}_i\}_{i=1}^K$ from the initialization $(\boldsymbol{z}_1, \boldsymbol{z}_2, ...)$ via the following PF-ODE:

$$\frac{d\boldsymbol{p}}{dt} = -\sigma(t)\dot{\sigma}(t)\mathbf{\Phi}_\theta(\boldsymbol{p}, t | O_t) \tag{13}$$

6:     $\hat{\boldsymbol{p}}_{\tau_1} \succ \hat{\boldsymbol{p}}_{\tau_2} ... \succ \hat{\boldsymbol{p}}_{\tau_K}$ where $\hat{\boldsymbol{p}}_{\tau_i} \succ \hat{\boldsymbol{p}}_{\tau_j} \iff \mathbf{\Psi}_\phi(\hat{\boldsymbol{p}}_{\tau_i}, \epsilon | O) > \mathbf{\Psi}_\phi(\hat{\boldsymbol{p}}_{\tau_j}, \epsilon | O)$        ▷ Ranking
7:     Estimate the current pose $\boldsymbol{p}_t = (\frac{\sum_{i=1}^M \hat{T}_{\tau_i}}{M}, \hat{\boldsymbol{q}}_t)$ where $M = \lfloor \delta \cdot K \rfloor$ and:

$$\hat{\boldsymbol{q}}_t = \arg\max_{\boldsymbol{q} \in SO(3)} \boldsymbol{q}^T \left( \frac{\sum_{i=1}^M A(\hat{\boldsymbol{q}}_{\tau_i})}{M} \right) \boldsymbol{q}, \; A(\hat{\boldsymbol{q}}_{\tau_i}) = \hat{\boldsymbol{q}}_{\tau_i}\hat{\boldsymbol{q}}_{\tau_i}^T \tag{14}$$

8: **end for**

---

# E  Ethics Statement and Boarder Impact

Our method has the potential to develop the home-assisting robot, thus contributing to social welfare. We evaluate our method in synthesized or human-collected datasets, which may introduce data bias. However, similar studies also have such general concerns. We do not see any possible major harm in our study.

