# OpenReview forum: "Generative Category-level Object Pose Estimation via Diffusion Models"
_NeurIPS.cc/2023/Conference — NeurIPS 2023 poster_

### Official Review · Reviewer_sAr7 · 2023-06-23

**Soundness:** 3 good
**Presentation:** 4 excellent
**Contribution:** 3 good
**Rating:** 7
**Confidence:** 4

**Summary:**

This paper proposes a novel approach for generative object pose estimation based on diffusion models.


**Strengths:**

1. Formulating pose estimation as a diffusion process is novel. It shows excellent ability to solve the multi-hypothesis issues in pose estimation caused by symmetry and partial observation. Moreover, it can be easily extended to solve the object pose tracking problem.

2. This paper proposes an efficient framework to estimate the object pose using diffusion models. It trains a score-based diffusion model to sample pose hypotheses, and an additional energy-based diffusion model to estimate the likehood of each hypothesis.

3. The paper is well written with nice figures, clear equations and proficient writing style.

4. Experimental results on several benchmarks demonstrate the efficiency of the proposed method.

**Weaknesses:**

In Table-1, this paper conducts comparison with category-level object pose estimation methods that predict 9 DoF object pose, consisting of 3D rotation, 3D translation and 3D size. However, the 3D size is not considered by the proposed method. Though the metrics are focused on rotation and translation, it still could cause a potential unfair comparison.


**Questions:**

As a likehood is estimated for each pose hypothesis, maybe an alternative way to get the final estimation is fusing them with a weighting scheme. For example, could it be possible to design a weighted version of Eq-9?

**Limitations:**

The limitation of inference speed and future work is included in the last section of the paper.

---

> ### Author Rebuttal · Authors · 2023-08-10
>
> > **(6D -> 9D)Q1: In Table-1, this paper conducts comparison with category-level object pose estimation methods that predict 9 DoF object pose, consisting of 3D rotation, 3D translation and 3D size. However, the 3D size is not considered by the proposed method. Though the metrics are focused on rotation and translation, it still could cause a potential unfair comparison.**
>
> **A1:**  Thanks for pointing it out! To clarify, although our method is primarily geared towards 6D object pose estimation, it can directly produce a 9D object pose when provided with a point cloud and segmentation mask. We first map the object point cloud to canonical space using its estimated 6D pose. Then, we refine the canonical point cloud using a standard outlier removal algorithm. Finally, we determine the 3D scales by computing the axis-aligned bounding box of the refined point cloud. The 9D object pose computation process is deployed in the real-world experiments found on our project page and in our supplementary video. We understand the importance of estimating 3D scales and are open to offering a more exhaustive evaluation if needed.
>
>
> > **(Mean Pooling v.s. Weighted Mean Pooling)Q2: As a likehood is estimated for each pose hypothesis, maybe an alternative way to get the final estimation is fusing them with a weighting scheme. For example, could it be possible to design a weighted version of Eq-9?**
>
> **A2:** Thanks for your insightful suggestions! We concede that there might be other aggregation techniques superior to simply averaging. Nonetheless, the primary concern in this work is dealing with outliers. The presence of outliers invariably skews aggregated results, regardless of the chosen method, be it weighed Mean Pooling or Mean Pooling. While some research addresses the multi-hypothesis challenge, they overlook this foundational problem. To our best knowledge, this is the first work that leverages energy-based diffusion model to remove outliers, which is also the key technical novelty of this work.
>
> Nevertheless, we conducted experiments to compare mean-pooling with other aggregation methods (i.e., likelihood weighting). As shown in **Table 1** of the PDF, mean-pooling consistently outperforms weighted averaging. We hypothesize that the energy model is better at distinguishing poses with significant differences in errors but performs poorly when distinguishing poses with low errors. As a result, some poses with higher errors might overshadow those with lower errors in energy weighting. To validate this hypothesis, we further explored the relationship between the pose error and the energy output. Results in **Figure 1** of the PDF demonstrate that the energy model assigns similar (right, <2 cm), or even incorrect values (left, <10 degrees) for poses with low errors. Meanwhile, the energies of low-error poses (e.g., <10 degrees, <2 cm) are higher than those of high-error poses (e.g., >20 degrees, >4 cm).

---

> > ### Comment · Reviewer_sAr7 · 2023-08-18
> > **Post-rebuttal comment**
> >
> > I appreciate the author's feedback, which addressed my concerns about how object size can be estimated on novel shapes, and how different aggregation scheme would perform with the proposed framework.
> >
> > Overall, I agree with other reviewers that this paper has proposed an efficient generative object pose estimation method based on diffusion models with sufficient novelty to be accepted by NeurIPS.

---

> > > ### Author Response · Authors · 2023-08-18
> > > **Thank You!**
> > >
> > > We are so glad that our responses help address your concerns. Thanks again for all your valuable feedback!

---

### Official Review · Reviewer_sHLy · 2023-06-30

**Soundness:** 3 good
**Presentation:** 3 good
**Contribution:** 3 good
**Rating:** 6
**Confidence:** 4

**Summary:**

This paper proposes a novel conditional generative approach for category-level pose estimation to solve the multi-hypothesis problem. The proposed method utilize the energy-based diffusion model to aggregate the candidates generated by the score-based diffusion model. Extensive experiments have been conducted to show the effectiveness of the proposed method on the REAL275 dataset under category-level pose estimation/tracking and novel category pose estimation evaluation protocols.

**Strengths:**

1. The author states the method clearly, and the notation is easy to follow.
2. The author proposed a novel multi-hypothesis method and shows promising results compared to regression and correspondence methods.
3. The ablation study is convincing.

**Weaknesses:**

1. I wonder if scoreNet and energyNet are needed for each class. Does a single model infer all classes?
2. I wonder about the inference time of object pose estimation. Is the object pose estimation inference time linearly increased according to the number of hypotheses?
3. Is there anything to consider about symmetries properties when augmentation in the training process?

**Questions:**

1. If it is not the model that predicts all classes, but each model, how did you compare the parameters of the model (Table 1)?

**Limitations:**

Covered the limitation in the main paper.

---

> ### Author Rebuttal · Authors · 2023-08-10
>
> > **Q1: I wonder if scoreNet and energyNet are needed for each class. Does a single model infer all classes?**
>
> **A1:** Apologies for the confusion. To clarify, we only require one set of both the score and energy models for all classes. Notably, neither model is conditioned on the class label or categorical priors, such as a mean point cloud. We will revise the paper to make this point clearer.
>
>
> > **Q2:  I wonder the inference time of object pose estimation. Is the object pose estimation inference time linearly increased according to the number of hypotheses?**
>
> **A2:** Thank you for raising this point. We concur that inference time is a critical concern. Due to the time-consuming sampling process of our diffusion model, our model achieves only 3FPS when handling the 6D object pose estimation based on single-frame images. However, to counter this challenge, we introduced a tracking algorithm. By leveraging well-initialized values from the previous frame, we significantly accelerate the sampling process, enabling the tracking frame rate to reach **18FPS**. Furthermore, we are actively exploring avenues to expedite the sampling process [1][2], marking an area of our future work.
>
> The inference time for our single-frame object pose estimation doesn't scale linearly with the number of pose hypotheses. This is because the hypotheses are sampled in batch-wise parallel (on the GPU) using the reverse ODE process.  Below, we list the inference times for varying numbers of pose hypotheses:
>
> | | 1 | 10 | 20 | 30 | 40 | 50 |
> |---|---|---|---|---|---|---|
> |   Single Frame Pose Estimation (s) |  0.231 |   0.285 |   0.292   | 0.301 | 0.316 | 0.328 |
> |               Pose Tracking (s)               | 0.036 | 0.043   |  0.046	 |0.049	 | 0.052	 | 0.054 |
>
> [1] Yang Song, Prafulla Dhariwal, Mark Chen, and Ilya Sutskever. Consistency models. ICML 2023
> [2] Tim Salimans and Jonathan Ho. Progressive distillation for fast sampling of diffusion models. ICLR 2022.
>
> > **Q3: Is there any symmetric augmentation in the training process?**
>
> **A3:** To clarify, we didn't incorporate specific designs, like tailored loss functions or symmetry-focused data augmentation, for symmetrical objects during training. Even so, we noted significant performance improvements, particularly with symmetric objects (as detailed in Sec 4.4 of the main text). We credit this success to our conditional generation formulation, which we believe provides valuable insights for the wider research community.
>
> > **Q4: If it is not the model that predicts all classes, but each model, how did you compare the parameters of the model (Table 1)**
>
> **A4:** As mentioned in **Q1**, we only train a single model to infer all categories in our paper.
>
> **We hope our rebuttal could address your concerns. Please let us know whether you have further questions. We are sincerely waiting for discussion with you.**

---

> > ### Comment · Reviewer_sHLy · 2023-08-17
> > **Re: Rebuttal by Authors**
> >
> > Thanks for your comments and most of my concerns are addressed. I have a follow-up question.
> >
> > 1. As I understood proposed method work without conditioning the class label or priors and is not designed for specific class property (symmetries). It looks like It can be worked for training with bowl, bottle, and can classes since their point cloud has similar geometry shapes. If we consider all training classes including laptop and camera, and mug, I wonder if the overall performance of the bowl, and bottle, can are maintained or degraded.

---

> > > ### Author Response · Authors · 2023-08-18
> > >
> > > Thank you for your response. We believe that your major concern is the scalability of our approach which is trained on all the categories without being conditioned on canonical priors if we understood correctly. To address it:
> > >
> > >  - Firstly, our model has already been trained on all six categories and achieved SOTA performance, as illustrated in Table 1 in the manuscript.
> > >
> > >  - Furthermore, as you mentioned, we conducted additional ablation studies on training categories. However, the training of *Ours w/o bottle, bowl, and can* could not converge within the discussion period due to limited time. Alternatively, we excluded one category from 'bottle,' 'bowl,' or 'can' during training and reported the average performance on the 'camera,' 'laptop,' and 'mug' categories.
> > > The results below demonstrate that the addition of a new category during training (referred to as *Ours w all categories*) **does not have a significant impact on the average performance of the existing categories** (i.e., 'camera,' 'laptop,' and 'mug'). This suggests the potential of our method to scale up to larger datasets.
> > >
> > > ***
> > >
> > > | Training categories | $5^{\circ}2$cm | $5^{\circ}5$cm | $10^{\circ}2$cm | $10^{\circ}5$cm |
> > > |---|---|---|---|---|
> > > |  *Ours w/o bottle*       |      31.27      |      42.73      |      55.80      |        71.73      |
> > > |  *Ours w/o bowl*        |      30.23      |      41.37      |      54.07      |        69.33      |
> > > |  *Ours w/o can*          |      32.23      |      43.00      |      55.23      |        70.07      |
> > > |  *Ours w all categories*  |      31.60      |      42.40      |      55.77      |        71.13      |
> > >
> > > ***
> > >
> > > We remain open to further discussions and inquiries on this concern and are sincerely waiting for your response!

---

> > > > ### Comment · Reviewer_sHLy · 2023-08-19
> > > > **Re: Official Comment by Authors**
> > > >
> > > > Thank you for considering all categories of experiments. My concerns have been addressed. I wonder If the proposed methods are scalable. Do you think it can be applied to general object pose estimation if your method is trained from large poses and various class amounts of datasets? Or are there any bottlenecks in being scalable to large scales?

---

> > > > > ### Author Response · Authors · 2023-08-19
> > > > >
> > > > > Thank you for bringing up this insightful question! We believe that our method could be scalable on a large scale. **Diffusion-based approaches have already demonstrated strong scalability in various conditional generation tasks**, such as image generation [1]. Notably, StableDiffusion [1], a well-established diffusion-based image generation model, has produced impressive results on ImageNet [4] and MS-COCO [5], both of which contain a vast array of conditions, such as categories and texts. In light of this, we believe that our diffusion-based approach can also be scalable in domains with significantly fewer dimensions than images.
> > > > >
> > > > > Regarding the potential "bottleneck", we would like to note that **diffusion models require a considerable amount of training time** [2,3]. In our experiments, we trained ScoreNet on a single RTX3090 for a period of two weeks. Given this, scaling our method to larger datasets may necessitate a greater amount of computational resources.
> > > > >
> > > > > We hope that our response adequately addresses your concerns. **If we've covered your concerns, we would greatly appreciate it if you would consider raising your score**. Thanks again for your valuable feedback!
> > > > >
> > > > >
> > > > > [1] High-Resolution Image Synthesis with Latent Diffusion Models, CVPR 2022
> > > > >
> > > > > [2] Generative Modeling by Estimating Gradients of the Data Distribution, NeurIPS 2020
> > > > >
> > > > > [3] Score-based Generative Modeling Through Stochastic Differential Equations, ICLR 2021
> > > > >
> > > > > [4] Imagenet: A large-scale hierarchical image database, CVPR 2009
> > > > >
> > > > > [5] Microsoft COCO: Common Objects in Context, ECCV 2014

---

> > > > > ### Author Response · Authors · 2023-08-21
> > > > > **We are looking forward to your reply!**
> > > > >
> > > > > As the author-reviewer discussion is ending, we would like to ask whether our replies have addressed your concerns. Please let us know whether you have further questions. We are sincerely waiting for your response!

---

### Official Review · Reviewer_W8Ay · 2023-07-06

**Soundness:** 2 fair
**Presentation:** 2 fair
**Contribution:** 3 good
**Rating:** 5
**Confidence:** 4

**Summary:**

This paper mines and formulates ambiguity in the task of object pose estimation, proposing to use a diffuse generative model to generate multiple hypotheses, which are then aggregated through additional scoring and ranking by another scorer net. The model achieves significant improvements with less supervision and parameters, has good cross-category generalization results, and performs well on tracking and stimulation tasks.

**Strengths:**

- I like the observation of the symmetric ambiguity and generative multi-hypothesis formulation.
- The improvement has been dramatically boosted, and it can beat all SOTAs even with fewer supervision signals and parameters.
- Thank the authors for testing the method on many datasets and tasks.

**Weaknesses:**

- (Conditional generation) Adding objects as conditions should further improve in-distribution test performance, right?
(Redundancy of model designs) EnergyNet can also be used for sampling, so why is ScoreNet needed?
- (Model design choices) How about using other models instead of EnergyNet to estimate, e.g., normalizing flows can give exact likelihood? There has been much multi-hypothesis work in human pose estimation/motion generation [a-c]. Essentially, this framework does not explore or mine features for object pose estimation. In other words, this model can also be used in other fields, and models in other fields can also be used in this field. Regarding the problem setup, there seem to be no many special things about monocular estimation except the mentioned symmetric ambiguity; however, the dramatic improvements in straightforwardly applying generative models encourage us to think about the underlying principles.
- (Mean pooling aggregation) Why must mean pooling to be used for aggregation? It looks like the mode of the largest cluster corresponds to the most likely solution, which is probably better than the mean. Additionally, some other works also study empirical aggregations, e.g., clustering [a], weighting [b] and [c]. Is it because quaternions are not easy to work on? What about converting to other forms (e.g., coordinates)? I expect to see comparisons or authors’ discussions on this.
- (Less yet better?) Can you elaborate on your speculation for the L265-267? Why does your method require less supervision information and a lighter model yet achieve better results? This guess will be very interesting.
- (Best hypothesis) I don't quite understand the last line in the table in Tab. 3; why is mean pooling needed since GT is accessible? What did I miss?
- (Cross-category generalization) I think the authors should emphasize the limitations when claiming generalization gains. I don't particularly understand that even the generative model also has unstable, bad, and unreliable performance for OOD point cloud conditions that have not been seen in the training set. And comparing the baselines in Tab. 5 is unfair; it is better to let them also train and test on the same split. I hope the author could kindly elaborate on this.
- (Missing related works) Should add related work of multi-hypothesis generation [a-e].
- (Visualizations of multiple pose hypotheses) While there is a dimensionality reduction visualization in Fig. 3, it is suggested to add the predicted multiple pose hypotheses along with the estimated energies to illustrate the method's effectiveness qualitatively. I also hope that the author can provide visualizations of the results, showing the advantages and failures of your method compared to the baseline under different settings such as D and RGB-D (Sup. Fig. 2 is better to also show input depth and point cloud data). This helps readers better understand the proposed method.
- (Reproducibility) While this cannot be forced or required, if the author commits to publicizing the code, it will have a greater impact and be more helpful to the field.

References:
- [a] C. Li & G. Lee. Weakly supervised generative network for multiple 3D human pose hypotheses. BMVC’20.
- [b] B. Biggs et al. 3D multibodies: Fitting sets of plausible 3D models to ambiguous image data. NeurIPS’20.
- [c] W. Shan et al. Diffusion-based 3D human pose estimation with multi-hypothesis aggregation. CVPR’23.
- [d] G. Chliveros et al. Robust multi-hypothesis 3D object pose tracking. ICVS’13.
- [e] F. Michel et al. Global hypothesis generation for 6D object pose estimation. CVPR’17.

**Questions:**

See Weaknesses.

**Limitations:**

Yeah.

---

> ### Author Rebuttal · Authors · 2023-08-10
>
> > **Q1: why is mean pooling needed since GT is accessible? What did I miss?**
>
> **A1:** To clarify, the ground truth (GT) is not accessible during test time. That's why we employ another energy-based diffusion model to aggregate these candidates into a final output in the absence of the GT.
>
> In our experiments, we emphasize the performance without the GT label in Table 1 of the main text. Additionally, we assess the upper limit of our approach by evaluating the best candidate (i.e., the one closest to the GT), in line with what other multi-hypothesis studies do[1].
>
> > **Q2: EnergyNet can also be used for sampling, so why is ScoreNet needed?**
>
> **A2:** Thanks for the good question. There are two main reasons:
>  - The score-based model significantly outperforms the energy-based model in candidate generation. We conducted an ablation study to compare the performance of using the score and energy models for candidate generation, respectively. As shown in **Table 2** of the PDF, 'score + energy' markedly outperforms 'energy + energy' across all metrics. This might be due to the limited capabilities of using the 2nd-order derivatives of an energy model to parameterize the score function.
>
>  - Furthermore, using the energy-based model to sample candidates requires calculating the second-order derivatives, which is time-inefficient.
>
> > **Q3: Why must mean pooling to be used for aggregation?**
>
> **A3:** Thank you for your valuable suggestion. Due to the character limit, please refer to **Q3** in the Common Responses.
>
> > **Q4: Why does your method require less supervision information and a lighter model yet achieve better results?**
>
> **A4:** Thank you for highlighting this interesting point. We hypothesize that the performance gain arises from the enlarged computational graph induced by the denoising process. During inference, the pose candidates are generated from a denoising process that involves hundreds of inferences from the score network. Although the score network has only **2.2M** parameters, the expanded computational graph due to the denoising process can grow to encompass hundreds of millions of parameters.
>
> > **Q5: This framework does not explore or mine features for object pose estimation. What is the underlying principles of the dramatic improvements?**
>
> **A5:** Thank you for bring this up. Sorry for any confusion regarding our unique design features.
>  - (Special Design) Due to the character limit, please refer to **Q1** in Common Response.
>
>  - (Underlying Priciples) Moreover, we concur that understanding the principles behind the significant improvements is valuable. We hypothesize that these improvements might stem from the substantially expanded computational graph of the denoising process and the energy model's capacity to eliminate outliers. We delve into these hypotheses in **Q3** and **Q4**, respectively. We will incorporate these analyses in our subsequent revision.
>
> > **Q6: The authors should emphasize the limitations when claiming generalization gains. Comparing the baselines in Tab. 5 is unfair**
>
> **A6:** We apologize for the oversight in clarifying the limitations. The limitations in cross-category generalization stem from conditional generation, as statistical methods can't guarantee out-of-distribution (OOD) generalization. For instance, if trained on REAL275 without cameras, our method may not perform well on camera objects due to their distinct geometry. We'll update Sections 4.4 and 5 to highlight these limitations more clearly.
>
> We are still retraining all the baselines using the same split as ours (see Table 5 of the PDF). We will update the final results as soon as possible during the discussion period.
>
> > **Q7: Adding objects as conditions should further improve in-distribution test performance, right?**
>
> **A7:** We agree that adding such prior (e.g., a 'mean point cloud') as input conditions might further improve the in-distribution test performance, which is also observed by previous caninocal-based methods.
> Nonetheless, such design choice would also hurt the OOD performance when encountered with objects from novel categories, due to the large shape variations between the seen and unseen categories.
>
> > **Q8: How about using other models instead of EnergyNet to estimate likelihoods, e.g., normalizing flows?**
>
> **A8:** Thanks for the good question. We agree that flow-based model could also be an alternative approach for exact likelihood estimation. Nevertheless, normalizing flows have to use specialized architecture to build a normalized probability model, which limits their capability. Besides, estimating exact likelihood from normalizing flows requires calculating the determinant of Jacobian matrix of z to x mapping, which is time-consuming. On the other side, diffusion-based methods have achieved SOTA performance on NLL test[2].
> In light of this, we choose to use EnergyNet to estimate the likelihood. We will include the rationale of choosing the likelihood estimator
>
> > **Q9: Should add related work of multi-hypothesis generation [a-e].**
>
> **A9:** Thanks for your advice! We will include these related works in the future revision.
>
> > **Q10: Add the predicted multiple pose hypotheses along with the estimated energies. Showing the advantages and failures of your method**
>
> **A10:** Thanks for the good suggestion! We promise to revise Fig.2 and Fig.3 accordingly and provide qualitative results of advatanges/failure cases in the future revision. However, due to the page limit of PDF, we do not demonstrate them at the rebuttal stage.
>
> > **Q11: Releasing the codes will have a greater impact and be more helpful to the field.**
>
> **A11:** We promise to release the code upon acceptance. We are also open to deploying our model on HuggingFace to validate its effectiveness, if required.
>
> [1] Shan et al. Diffusion-Based 3D Human Pose Estimation with Multi-Hypothesis Aggregation.
>
> [2] Song et al. Score-based generative modeling through stochastic differential equations.

---

> ### Author Response · Authors · 2023-08-16
> **Updating the Cross-category Experiments' Result**
>
> Apologize for the late update. We have finished the training and compared the performance of our method and the baselines on cross-category generalization, employing the same training and testing split, in the table below. Results show that our method still outperforms the baselines in the OOD test significantly.
>
> We hope this could address your concerns and **are sincerely waiting for your response.** If you have any further questions, please let us know and we will spare no effort to provide in-time responses.
>
> ***
> | Category | Method | $5^{\circ}2$cm | $5^{\circ}5$cm | $10^{\circ}2$cm | $10^{\circ}5$cm |
> |---|---|---|---|---|---|
> |             |  SAR-Net[1]  |      58.1/36.4      |      66.0/47.3      |      83.7/59.4      |        93.6/81.5      |
> |  bowl   | RBP-Pose[2] |      75.4/0.0        |      81.7/6.9        |      92.1/0.1        |       100.0/30.7     |
> |             |       Ours       | **85.7**/**64.5** | **92.6**/**72.5** | **93.1**/**87.2** | **100.0**/**98.6** |
> |---|---|---|---|---|---|
> |             |  SAR-Net[1]  |      43.5/11.7      |      54.0/23.0      |      61.3/33.6      |        79.8/68.0      |
> |  bottle  | RBP-Pose[2] |      38.7/4.3       |      43.5/5.8        |      76.4/24.7       |       89.8/29.7      |
> |             |       Ours       | **53.6**/**39.0** | **62.0**/**53.2** | **81.4**/**73.6** |  **92.7**/**94.6**  |
> |---|---|---|---|---|---|
> |             |  SAR-Net[1]  |      32.2/7.3        |      62.2/52.3      |      52.5/12.1      |        92.9/87.9      |
> |    can   | RBP-Pose[2] |      53.5/0.8        |     67.1/21.0       |      78.8/2.6        |        96.3/61.7      |
> |             |       Ours       | **73.2**/**62.5** | **81.2**/**74.0** | **88.8**/**81.6** |  **99.8**/**99.7**  |
>
> **Caption:** On the left side of the '/' are the results when all categories were included in the training, while on the right side of the '/' are the results when testing categories were excluded from training.
> ***
>
> Notably, as [1] and [2] requires category priors, we provide them with category prior from the nearest related category: When tested on  'bottle', 'bowl', and 'can', [1] and [2] are provided with 'can', 'bottle', and 'bottle' priors, respectively.
>
> [1] Lin H, et al. Sar-net: Shape alignment and recovery network for category-level 6d object pose and size estimation. CVPR, 2022.
>
> [2] Zhang R, et al. RBP-Pose: Residual bounding box projection for category-level pose estimation. ECCV, 2022.

---

> > ### Comment · Reviewer_W8Ay · 2023-08-19
> >
> > I sincerely thank the author for his meticulous response to my concerns, which resolved many of them. I tend to improve my rating. In addition, if we can discuss the remaining concerns, I think it will be more conducive to improving this work.
> >
> > (**Improvement understanding**) Personally, I am concerned about the explanation of diffusion models' extended computation graph. It would be even better if the author could provide some references to support it. I think the main takeaway from this work is the formulation of generative tasks, which allow generative models to model ambiguity well. Better generative models may lead to better performance. But as more conditions are added to reduce ambiguity, the advantage of generative models seems to diminish. I am looking forward to the authors' comments on my opinions.
> >
> > (**Different choices of EnergyNet**) I still think it would make this work more complete if the authors could consider comparing performance differences between the current diffusion EnergyNet and commonly used normalized flows in a future version (not necessarily at this discussion stage).
> >
> > (**Heuristic mean pooling**) I also want to comment on heuristic weighting. Thanks to the author for his in-depth research. The results show that EnergyNet is not well calibrated in rotation (i.e., lower errors should have higher energy). This is why more reasonable designs are not good, such as weighting pooling by energy. And this kind of trustworthy-related problem and phenomenon deserves attention and in-depth study.
> >
> > (**Better performance with less supervision**) Since the author also said in R7 that adding category priors can help improve the performance of in-distribution testing, can the author further guide me about the understanding of your better performance with less supervision (depth or category prior) compared to existing work (L265)? Is that because using EnergyNet helps remove outliers to make the prediction more accurate? If so, does that mean the benefits from removing outliers are larger than those from more supervision?
> >
> > (**Good OOD generalization understanding**) I still don't understand its rationale, such as why the model is not trained on the bowl category but can generalize well to the OOD bowl category. The in-distribution generalization MLE objective for generative model training does not account for the good generalization achieved.
> >
> > (**ScoreNet vs. EnergyNet**) If I understand the point in the paper [35] correctly, they say that it doesn't matter if the EBM is not normalized, but factors like architecture do. So that's why I don't understand your experimental results on the different generative performances of ScoreNet and EnergyNet with the same architecture in your case. Can you elaborate?

---

> > > ### Author Response · Authors · 2023-08-21
> > > **Reply to the followup concerns Part[1/2]**
> > >
> > > Thank you for providing us with your valuable and well-structured feedback, and for also improving the rating! We sincerely appreciate the in-depth and insightful discussion we've had with you and the other reviewers. Furthermore, we assure you that in future revisions, we will make an effort to incorporate additional results into the main text or appendix during the discussion period. **We firmly believe that your valuable suggestions and questions contribute significantly to the improvement of this work,** and we are more than willing to address your follow-up questions:
> > >
> > > ***
> > >
> > > > **Q1: (Improvement understanding)**:
> > >
> > > **A1:** Our explanation of the extended computational graph can be found in [1], specifically in the second paragraph of the Introduction. The first author of [1] is the pioneer of the deep score-based diffusion model [2, 3]. Furthermore, we have performed additional experiments to bolster this explanation. These results are presented in Table 4 of the PDF, where performance consistently improves with an increasing number of sampling steps.
> > >
> > > Additionally, we believe that the ambiguity arising from partial observations would persist, no matter how many conditions are added. For instance, if the handle of a cup remains invisible within the current view, the introduction of RGB images or canonical priors would not eliminate the pose ambiguity of the cup. Therefore, we are of the opinion that the inclusion of further conditions would not diminish the advantages of the generative models.
> > >
> > > [1] Consistency Models, ICML 2023
> > >
> > > [2] Generative Modeling by Estimating Gradients of the Data Distribution, NeurIPS 2020
> > >
> > > [3] Score-based Generative Modeling Through Stochastic Differential Equations, ICLR 2021
> > >
> > > > **Q2: (Better performance with less supervision)**:
> > >
> > > **A2:** Thanks for your insightful questions! We believe that our approach achieves improved performance with reduced supervision, not primarily due to energy-based outlier removal, but rather because of the effectiveness of our generative formulation in modeling ambiguities and the decision to utilize diffusion models. The results presented in Table 3 of our manuscript, using the *random ranker*, demonstrate that while energy-based outlier removal contributes to the enhanced performance of our method, it is not the primary factor driving the significant performance improvements we have achieved. Nonetheless, we recognize that leveraging additional sources of supervision, such as the RGB images or canonical priors, constitutes a promising future direction that aligns well with our current approach.
> > >
> > > > **Q3: (Good OOD generalization understanding)**:
> > >
> > > **A3:** We appreciate you bringing this to our attention. We hypothesize that the specific out-of-distribution (OOD) generalization ability of our method arises from the learned feature space of the point cloud. Although the bowl category is considered OOD, the extracted features from point clouds in the bowl category may exhibit similarities or closeness to seen categories, to some extent. In other words, a bowl may share visual characteristics with items such as cans, bottles, or mugs, as identified by the PointNet of the ScoreNet.
> > >
> > > **To validate this hypothesis, we conducted a t-SNE[4] analysis on the point cloud feature space of the ScoreNet.** Specifically, we employed the PointNet of the ScoreNet trained on five categories excluding bowls as a feature extractor. Subsequently, we extracted features from the point clouds of objects in the test set and visualized the t-SNE results. The outcomes of the t-SNE analysis (w/o bowl), along with representative CAD models of bowls, cans, and bottles, are depicted on the top of our anonymous website (accessible via the link provided in the abstract) within the manuscript. Additionally, we also include the same t-SNE test on w/o can and w/o bottle.
> > >
> > > The results demonstrate that features from cans and bottles tend to intermingle, aligning with the accurate observation that both cans and bottles exhibit symmetrical cylindrical shapes. Meanwhile, features from the bowl category show proximity to features from mugs. This phenomenon indicates a degree of similarity between bowls and other training categories within the feature space.
> > >
> > > [4] "Visualizing Data using t-SNE," JMLR 2008"

---

> > > > ### Author Response · Authors · 2023-08-21
> > > > **Reply to the followup concerns Part[2/2]**
> > > >
> > > > > **Q4: (ScoreNet vs. EnergyNet)**:
> > > >
> > > > **A4:** We would like to clarify that the performance gap between ScoreNet and EnergyNet arises from the difference in their parameterizations of the score function:
> > > >
> > > >  - For ScoreNet, the score function is parameterized as $\mathbf{\Phi}_{\theta}(p, t| O)$.
> > > >
> > > >  - In contrast, EnergyNet's score function is parameterized as:
> > > > $\nabla_{p}\mathbf{\Psi}{\phi}(p, t| O)$
> > > > $= \nabla_{p} \langle p, \mathbf{\Phi}{\phi}(p, t| O)\rangle$
> > > > $= \mathbf{\Phi}{\phi}(p, t| O)$
> > > > $+ \mathbf{p}^T \nabla_{p} \mathbf{\Phi}{\phi}(p, t| O)$
> > > >
> > > > Hence, while $\mathbf{\Phi}{\phi}(\mathbf{p}, t| O)$ and $\mathbf{\Phi}{\theta}(\mathbf{p}, t| O)$ share the same architecture, the score function derived from EnergyNet differs from ScoreNet's due to the additional term $\mathbf{p}^T \nabla_{\mathbf{p}} \mathbf{\Phi}{\phi}(\mathbf{p}, t| O)$. We feel sorry for the misunderstanding and will elaborate more clearly in future revision.
> > > >
> > > > > **Q5: (Heuristic mean pooling)**:
> > > >
> > > > **A5:**  We appreciate your valuable suggestion! We concur that opting for weighted mean pooling or clustering would indeed constitute a more rational approach. Furthermore, owing to the fact that the energy is not impeccably calibrated with errors, the transient advantages of weighted mean pooling may not be readily discernible. We wish to assure you that we are committed to including the energy error correlation analysis within Section 4 and comprehensively discussing the underlying rationale for heuristic pooling selections in Section 3.
> > > >
> > > >
> > > > > **Q6: (Different choices of EnergyNet)**:
> > > >
> > > > **A6:** Thank you for the good suggestion! We intend to incorporate a comparison between EnergyNet and the Flow-based method in our future revision. Specifically, we will follow [5] to implement the flow-based model for our task. Additionally, we will meticulously determine the network capacity of the flow-based network to ensure a fair comparison.
> > > >
> > > > [5]  B. Biggs et al. 3D multibodies: Fitting sets of plausible 3D models to ambiguous image data. NeurIPS’20.
> > > >
> > > > ***
> > > >
> > > > We hope that our response adequately addresses your concerns. **If we've covered your concerns, we would greatly appreciate it if you would consider further raising your score.** Thanks again for your valuable feedback!

---

> > > > > ### Author Response · Authors · 2023-08-21
> > > > > **Additional Quantitative Results for Q3 (Good OOD generalization understanding)**
> > > > >
> > > > > We have supplemented the t-SNE analysis with quantitative assessments of point cloud feature similarity to provide a more rigorous comparison. These assessments include the chamfer distance between the unseen category and other categories, as well as the intra-class deviation of the feature L2 distance within each category.
> > > > >
> > > > > As shown in Table 1, the features of the bowl are mostly similar to the features of the mug. The chamfer distance between the bowl's features and those of the mug (1.3) is smaller than the intra-class deviation of the mug (1.65). This suggests that the PointNet might 'recognize' some of the (unseen) bowl's point clouds as those of the mug, which aligns with the observations from the t-SNE results.
> > > > >
> > > > > **As the author-reviewer discussion is ending, we would like to ask whether our replies have addressed your concerns for updating your rating.** Please let us know whether you have further questions. We are sincerely waiting for your response.
> > > > >
> > > > > ***
> > > > >
> > > > > |                            | bottle | camera | can  | laptop | mug  |
> > > > > |----------------------------|--------|--------|------|--------|------|
> > > > > | chamfer distance with bowl | 1.91   | 1.4    | 1.39 | 1.71   | **1.3**  |
> > > > > | std                        | 1.66   | 1.52   | 1.77 | 2.06   | 1.65 |
> > > > >
> > > > > **Table 1**: Feature similarity comparison when training without *bowl* category.
> > > > > ***
> > > > >
> > > > > ***
> > > > >
> > > > > |                              | bowl | camera | can  | laptop | mug  |
> > > > > |------------------------------|------|--------|------|--------|------|
> > > > > | chamfer distance with bottle | 1.36 | 1.53   | **1.30** | 1.91   | 1.49 |
> > > > > | std                          | 1.45 | 1.58   | 1.81 | 2.17   | 1.81 |
> > > > >
> > > > > **Table 2**: Feature similarity comparison when training without *bottle* category.
> > > > > ***
> > > > >
> > > > >
> > > > >
> > > > > ***
> > > > >
> > > > > |                           | bottle | bowl | camera | laptop | mug   |
> > > > > |---------------------------|--------|------|--------|--------|-------|
> > > > > | chamfer distance with can | 1.75   | 1.72 | 1.8    | 1.76   | **1.32**  |
> > > > > | std                       | 1.43   | 1.29 | 1.42   | 1.98   | 1.54  |
> > > > >
> > > > > **Table 3**: Feature similarity comparison when training without *can* category.
> > > > >
> > > > > ***

---

### Official Review · Reviewer_4bwh · 2023-07-06

**Soundness:** 4 excellent
**Presentation:** 4 excellent
**Contribution:** 3 good
**Rating:** 7
**Confidence:** 4

**Summary:**

To settle the multi-hypothesis issue in category-level 6D pose estimation, this paper formulates the focused task as conditional generative modeling and proposes a novel method based on diffusion models, which utilizes a score-based diffusion model to sample pose candidates with an energy-based one followed to rank those candidates. The proposed method achieves the state-of-the-art results on REAL275 dataset for category-level pose estimation, and is further extended for the task of pose tracking.

**Strengths:**

- The authors propose a new perspective for category-level 6D pose estimation by formulating the task as conditional generative modeling and realizing it via diffusion models.

- The proposed method achieves the state-of-the-art results on REAL275 dataset for both tasks of category-level pose estimation and pose tracking. Cross-category experiments and real-world applications are also conducted to verify its generalization ability.

- The paper is well written and presented with detailed illustrations and thorough experiments.



**Weaknesses:**

- The proposed method is not competitive in terms of inference time with the use of generative models.

- The original definition of category-level task includes the estimation of object sizes, which could not be learned in the proposed method.

- Some methods are not compared with in Table 1 for pose estimation, e.g., [1][2][3], and Table 6 for pose tracking, e.g., [1][4][5].

Reference:

[1] Sparse Steerable Convolutions: An Efficient Learning of SE(3)-equivariant Features for Estimation and Tracking of Object Poses in 3d Space.

[2] CenterSnap: Single-Shot Multi-Object 3D Shape Reconstruction and Categorical 6D Pose and Size Estimation.

[3] ShAPO: Implicit Representations for Multi-Object Shape, Appearance and Pose Optimization.

[4] ICK-Track: A Category-Level 6-DoF Pose Tracker Using Inter-Frame Consistent Keypoints for Aerial Manipulation.

[5] BundleTrack: 6D Pose Tracking for Novel Objects without Instance or Category-Level 3D Models.



**Questions:**

- What’s the special designs of the proposed method for category-level pose estimation, compared to instance-level pose estimation? Could it be applied to the instance-level task?

- Some typos.
    - ’w‘ in Eq. (1) and $\dot{\sigma}$ in Eq. (4) are not explained.
    - In Line167, should '$log p_t(p|O)$' be '$\Delta_p log p_t(p|O)$'?
    - In Line213, "symmetric objects ??" -> "symmetric objects".
    - In Line278,281,284, should all the 'M' be K"?
    - In References, paper [12] and paper [13] refer to the same papers.

**Limitations:**

The authors have discussed the limitations of their work in the paper.

---

> ### Author Rebuttal · Authors · 2023-08-10
>
> > **Q1: The proposed method is not competitive in terms of inference time.**
>
> **A1:** Thanks for pointing it out! We acknowledge that the sampling process of the diffusion model does introduce a notable computational overhead. Consequently, our current approach for estimating 6D object pose from individual images does indeed lack competitive inference speed.To address this concern, we have demonstrated within the main body of our paper the adaptation of our method into a final framework for 6D object pose estimation, achieving a swift execution rate (18FPS) while maintaining high performance. Additionally, we are actively exploring a promising avenue for further research, which involves expediting the sampling process [1][2]. This area constitutes our future work.
>
> > **Q2: Object sizes could not be learned in the proposed method.**
>
> **A2:** Thanks for brining it up! To clarify, although our method is primarily geared towards 6D object pose estimation, it can directly produce a 9D object pose when provided with a point cloud and segmentation mask. We first map the object point cloud to canonical space using its estimated 6D pose. Then, we refine the canonical point cloud using a standard outlier removal algorithm. Finally, we determine the 3D scales by computing the axis-aligned bounding box of the refined point cloud. The 9D object pose computation process is deployed in the real-world experiments found on our project page and in our supplementary video. We understand the importance of estimating 3D scales and are open to offering a more exhaustive evaluation if needed.
>
>
> > **Q3: Some methods are not compared with in Table 1 for pose estimation, e.g., [1][2][3], and Table 6 for pose tracking, e.g., [1][4][5].**
>
> **A3:** We sincerely appreciate your valuable feedback, which has shed light on additional baseline methods. According to your suggestion, we have meticulously presented a comparative analysis between our proposed approach and the methods referenced in your comments.
>
> Regarding the primary task of category-level 6D object pose estimation, our method maintain a substantial lead in terms of performance:
> | Method | $5^{\circ}2$cm$\uparrow$ | $5^{\circ}5$cm$\uparrow$ | $10^{\circ}2$cm$\uparrow$ | $10^{\circ}5$cm$\uparrow$ |
> |---|---|---|---|---|
> |            [3]          |   36.6    |   43.4   |   52.6    |    63.5    |
> | CenterSnap[4]  |      -       |   29.1   |       -      |   64.3    |
> |     ShAPO[5]     |      -       |   48.8   |       -      |    66.8    |
> |          Ours        | **52.4** | **61.2** | **72.8**| **84.2** |
>
> For object tracking task, despite being directly transferred from a single-image prediction method, our approach has achieved comparable performance with the baselines:
> |Method| $5^{\circ}5$cm$\uparrow$ | $r_{error}\downarrow$ | $t_{error}\downarrow$ |
> |---|---|---|---|
> |                        [3]                         |    54.5   |   5.2    |   1.9    |
> |                ICK-Track[6]                 | **84.4** |   4.5    |   3.1    |
> | BundleTrack[8] w/o Pose Graph |    39.9   |   9.2    |   2.4    |
> |                      Ours                       |    71.5   | **4.2** | **1.5** |
> |              BundleTrack[7]               |    87.4   |   2.4    |   2.1    |
>
> Notably, the bundle track method achieved enhanced performance by employing multi-frame images for global optimization. However, it is noteworthy that this procedure holds generality. When subjected to a fair comparison utilizing only a pair of image frames, our approach demonstrates superiority over the bundle track method.
>
> We intend to include these particular baseline comparisons, as well as the references, in the revised version of our manuscript.
>
> > **Q4: What’s the special designs of the proposed method for category-level pose estimation, compared to instance-level pose estimation? Could it be applied to the instance-level task?**
>
> **A4:** Thank you for addressing this. For details on the special design, please refer to Q1 in the Common Responses. Our framework is specifically designed to address the multi-hypothesis challenge in category-level pose estimation. However, it can also be adapted for instance-level tasks by conditioning on the target object's CAD model. It's worth noting that this adapted method might not offer significant benefits over other instance-level methods.
>
> ### Thank you for the detailed response regarding typos. We will revise the paper accordingly. Below are the detailed explanations:
>
> > **T1: 'w' in Eq. (1) and \sigma in Eq. (4) are not explained.**
>
> **A1:** The $dw$ is the standard Wiener process[3] (a.k.a., Brownian motion). The $\dot{\sigma}(t)$ is the derivation of $\sigma(t)$: $\dot{\sigma}(t) = \sigma_{min} \cdot (\ln \sigma_{max} -\ln \sigma_{min}) (\frac{\sigma_{max}}{\sigma_{min}})^t$
>
> > **T2: In line 167, should `logp_t(p|O)' be \delta `log p_t(p|O)'?**
>
> **A2:** Yes, it shoud be $\delta_{p} \log p_t(p|O)$.
>
> > **T3: In Line278,281,284, should all the 'M' be K"?**
>
> **A3:** Yes! It should be $K$.
>
> [1] Song Y,  et al. Consistency models. ICML 2023.
>
> [2] Salimans T, et al. Progressive distillation for fast sampling of diffusion models. ICLR 2022.
>
> [3] Lin J, et al.  Sparse Steerable Convolutions: An Efficient Learning of SE(3)-equivariant Features for Estimation and Tracking of Object Poses in 3d Space. Advances in Neural Information Processing Systems, 2021.
>
> [4] Irshad M Z, et al. CenterSnap: Single-Shot Multi-Object 3D Shape Reconstruction and Categorical 6D Pose and Size Estimation. ICRA 2022.
>
> [5] Irshad M Z,  et al. ShAPO: Implicit Representations for Multi-Object Shape, Appearance and Pose Optimization. ECCV, 2022.
>
> [6] Sun J, et al. ICK-Track: A Category-Level 6-DoF Pose Tracker Using Inter-Frame Consistent Keypoints for Aerial Manipulation. IROS, 2022.
>
> [7] Wen B, et al. BundleTrack: 6D Pose Tracking for Novel Objects without Instance or Category-Level 3D Models. IROS, 2021.

---

### Official Review · Reviewer_7PCs · 2023-07-09

**Soundness:** 4 excellent
**Presentation:** 3 good
**Contribution:** 3 good
**Rating:** 7
**Confidence:** 4

**Summary:**

This paper proposes a diffusion-based model for category-level object pose estimation. Different from the existing deterministic approaches which treat the object pose estimation as a regression problem, the proposed diffusion model alternatively formulated it as a generation problem. In this way, it could tackle the multiple pose hypotheses issue comes from symmetric objects and partial observation.  The proposed approach achieves state-of-the-art performance on existing benchmarks and could generalize to unseen symmetric categories.

**Strengths:**

a. The paper introduces a novel diffusion-based framework for category-level pose estimation. The framework consists of two diffusion models. The first diffusion model generates a set of pose candidates for a given point cloud during the inference stage. The second diffusion model uses an energy-based approach to rank the candidates and filter out those with low rankings. This paper claims to be the first to propose a solution for pose estimation using diffusion models.

b. The diffusion-based model presented in the paper demonstrates state-of-the-art performance on existing benchmarks. It surpasses the performance of previous approaches by a significant margin. Additionally, the authors argue that their framework has the potential to generalize to objects of unseen categories that have not been studied by other approaches.

**Weaknesses:**

a. The chosen representation of the pose parameter in this paper consists of a 9-dimensional vector, where 6 dimensions are allocated for rotation and 3 dimensions for translation. Notably, the omission of the scale parameter from this representation raises an important inquiry regarding the reasons behind its exclusion and the means by which it can be obtained during inference. Further clarification is required to address these concerns.

b. The current approach employed by the authors involves utilizing mean pooling to derive the final pose estimation from the filtered pose candidates. Given that the energy-based diffusion model is capable of estimating the data likelihood for each pose candidate, an alternative consideration arises: would adopting weighted averaging be a more suitable approach for calculating the final pose estimation? It would be insightful to explore the potential benefits of incorporating weighted averaging as a means to enhance the accuracy of the pose estimation.

c. Additional information clarifying the specifics of the pose tracking implementation would be appreciated. Considering that the framework incorporates two diffusion models, it appears intriguing for me to comprehend how the pose tracking component can maintain such computational efficiency and deliver results in real-time.

d. Some typos. (Line 213)

**Questions:**

Some questions are listed in the weakness section.

**Limitations:**

The authors have discussed the limitations of this work.

---

> ### Author Rebuttal · Authors · 2023-08-10
>
> > **Q1: inquiry regarding the reasons about exclusion of scale parameters and the means by which it can be obtained during inference.**
>
> **A1:** Thanks for pointing this out! We would like to clarify that our work primarily focus on the estimation of a 6D object pose. Nonetheless, our method possesses the capability to derive a 9D object pose given the point cloud and the segmentation mask. This process involves several key steps:
>  - Transforming the object's point cloud into the canonical space, utilizing estimated 6D object pose.
>  - Denoising the caninocal point cloud via an off-the-shelf outlier removal algorithm.
>  - Calculating the axis-aligned bounding box of the denoised point cloud to get the object size.
>
> And in our real world experiments (can be seen on the project page or supplementary video), the 9D object pose is calculated via the aforementioned method and be used for robotic manipulation. We acknowledge that estimating 3D sizes is also important and willing to provide more comprehensive evaluations if required.
>
> > **Q2: would adopting weighted averaging be a more suitable approach for aggregation?**
>
> **A2:** Thanks for your insightful suggestion! We concede that there might be other aggregation techniques superior to simply averaging. Nonetheless, the primary concern in this work is dealing with outliers. The presence of outliers invariably skews aggregated results, regardless of the chosen method, be it K-means or Mean Pooling. While some research addresses the multi-hypothesis challenge, they overlook this foundational problem. To our best knowledge, this is the first work that leverages energy-based diffusion model to remove outliers, which is also the key technical novelty of this work.
>
> We conducted experiments to compare mean-pooling with other aggregation methods (i.e., likelihood weighting). As shown in **Table 1** of the PDF, mean-pooling consistently outperforms weighted averaging. We hypothesize that the energy model is better at distinguishing poses with significant differences in errors but performs poorly when distinguishing poses with low errors. As a result, some poses with higher errors might overshadow those with lower errors in energy weighting. To validate this hypothesis, we further explored the relationship between the pose error and the energy output. Results in **Figure 1** of the PDF demonstrate that the energy model assigns similar (right, <2 cm), or even incorrect values (left, <10 degrees) for poses with low errors. Meanwhile, the energies of low-error poses (e.g., <10 degrees, <2 cm) are higher than those of high-error poses (e.g., >20 degrees, >4 cm).
>
>
> > **Q3: clarifying the specifics of the pose tracking implementation.**
>
> **A3:**
> We apologize for any confusion caused. We have summarized our tracking algorithm in **Algorithm 1** of the PDF for your reference. Our algorithm is adept at maintaining both computational efficiency and high performance. This is primarily attributed to the benefit of warm-starting from previous predictions. Since the difference in object positions between adjacent frames is typically minimal, the ODE process requires fewer time steps to arrive at a reliable estimation from the previous one, thereby boosting efficiency.
>
> > **Q4: Some typos. (Line 213)**
>
> **A4:** Thanks for pointing out the typos, and we will address all the typos in the revised version.

---

> > ### Comment · Reviewer_7PCs · 2023-08-18
> > **Re: Rebuttal by Authors**
> >
> > Thanks for your feedback which addressed most of my concerns. After reviewing the comments from other reviewers,  I think this paper proposes a diffusion-based method for 6D pose estimation is quite novel and effective. I will raise my original rating.

---

> > > ### Author Response · Authors · 2023-08-18
> > > **Thank You!**
> > >
> > > Thanks for raising your rating to 7. We are so glad that our responses help address your concerns. Thanks again for all your valuable feedback!

---

### Author Rebuttal · Authors · 2023-08-10

We are grateful to all reviewers for recognizing the merit of our idea, experiments, and presentation:
 - "I appreciate the observation of symmetric ambiguity and the generative multi-hypothesis formulation." (W8Ay)
 - "The improvement has seen a significant boost." (W8Ay)
 - "This surpasses the performance of prior approaches by a considerable margin." (7PCs)
 - "We've also conducted cross-category experiments and real-world applications to validate its generalizability." (4bwh)
 - "The paper is meticulously written with comprehensive illustrations and thorough experiments." (4bwh)

**(Q1: Special Design)** However, we notice that some reviewers may be confused about our special design on object estimation tasks (**4bwh**, **W8Ay**). In this work, we address the multi-hypothesis issue, which is one of the **key features** of monocular object pose estimation. Moreover, to eliminate outliers resulting from conditional generation, our **key design** feature is the use of an energy-based diffusion model to filter them out. To highlight this primary objective, our networks consist solely of commonly used architectures. This aspect notably paves the way for future explorations of specialized architectural designs within our framework. We will clarify this further in our revised introduction.

| | Previous works | This work |
| --- | --- | ---|
| Training Paradigm | Regression-based | Conditional Generative Modeling|
| Key Challenge |  Handling Intra-class Variations |  Removing Outliers |

**(Q2: 3D Scale Estimation)** We also observed feedback from reviewers noting our omission of 3D scale estimation in the primary text (**7PCs**, **4bwh**, **sAr7**). To clarify, although our method is primarily geared towards 6D object pose estimation, it can directly produce a 9D object pose when provided with a point cloud and segmentation mask. We first map the object point cloud to canonical space using its estimated 6D pose. Then, we refine the canonical point cloud using a standard outlier removal algorithm. Finally, we determine the 3D scales by computing the axis-aligned bounding box of the refined point cloud. The 9D object pose computation process is deployed in the real-world experiments found on our project page and in our supplementary video. We understand the importance of estimating 3D scales and are open to offering a more exhaustive evaluation if needed.

**(Q3: Alternative Aggregation Methods)** Additionally, some reviewers have suggested alternative aggregation methods like weighted mean-pooling (**7PCs**, **W8Ay**, **sAr7**). We concede that there might be other aggregation techniques superior to simply averaging. Nonetheless, the primary concern in this work is dealing with outliers. The presence of outliers invariably skews aggregated results, regardless of the chosen method, be it K-means or Mean Pooling. While some research addresses the multi-hypothesis challenge, they overlook this foundational problem. To our best knowledge, this is the first work that leverages energy-based diffusion model to remove outliers, which is also the key technical novelty of this work.

To address reviewers' concerns, we conducted experiments to compare mean-pooling with other aggregation methods (i.e., likelihood weighting). As shown in **Table 1** of the PDF, mean-pooling consistently outperforms weighted averaging. We hypothesize that the energy model is better at distinguishing poses with significant differences in errors but performs poorly when distinguishing poses with low errors. As a result, some poses with higher errors might overshadow those with lower errors in energy weighting. To validate this hypothesis, we further explored the relationship between the pose error and the energy output. Results in **Figure 1** of the PDF demonstrate that the energy model assigns similar (right, <2 cm), or even incorrect values (left, <10 degrees) for poses with low errors. Meanwhile, the energies of low-error poses (e.g., <10 degrees, <2 cm) are higher than those of high-error poses (e.g., >20 degrees, >4 cm).

We sincerely hope our work contributes to the Machine Learning + 3D Vision research community. Below we reply to reviewers’ questions point-by-point. Thanks again for your valuable comments and suggestions!

---

> ### Comment · Reviewer_W8Ay · 2023-08-19
> **Add STD of Calibration in Reb. Fig. 1**
>
> **Q3:** Figure 1 (energy error correlation analysis) is appreciated. I suggest you include the standard deviation in addition to the mean. It will help to know more about calibration.

---

> > ### Author Response · Authors · 2023-08-21
> >
> > Your suggestion is greatly appreciated. We will be incorporating not only the mean but also the standard deviation in the upcoming version. Thank you for your input!

---

### Decision · Program_Chairs · 2023-09-21

**Decision:**

Accept (poster)

**Comment:**

This paper proposes a method to apply diffusion models to category-level object pose estimation. Experimental results show that the proposed method outperforms state-of-the-art methods. All reviewers acknowledged the novelty, clarity, and promising experimental results. In the weaknesses section, several concerns were raised regarding inference time and limitations of object size estimation, which were addressed in the authors' rebuttal. As a result, several reviewers raised their final scores, and we reached a consensus to accept the paper. It is strongly recommended that the final version of the paper include the discussion and experimental results provided in the rebuttal.